# Extracellular Vesicles Derived from Endothelial Progenitor Cells Protect Human Glomerular Endothelial Cells and Podocytes from Complement- and Cytokine-Mediated Injury

**DOI:** 10.3390/cells10071675

**Published:** 2021-07-02

**Authors:** Davide Medica, Rossana Franzin, Alessandra Stasi, Giuseppe Castellano, Massimiliano Migliori, Vincenzo Panichi, Federico Figliolini, Loreto Gesualdo, Giovanni Camussi, Vincenzo Cantaluppi

**Affiliations:** 1Department of Translational Medicine, University of Piemonte Orientale (UPO), via Solaroli 17, 28100 Novara, Italy; davidemedica@gmail.com; 2Nephrology, Dialysis and Transplantation Unit, Department of Emergency and Organ Transplantation, University of Bari, 70124 Bari, Italy; rossanafranzin@hotmail.it (R.F.); alessandra.stasi@uniba.it (A.S.); loreto.gesualdo@uniba.it (L.G.); 3Nephrology, Dialysis and Transplantation Unit, Department of Medical and Surgical Sciences, University of Foggia, 71122 Foggia, Italy; castellanogiuseppe74@gmail.com; 4Nephrology and Dialysis Unit, Versilia Hospital, 55041 Camaiore, Italy; maxmigliori@yahoo.it (M.M.); vincenzo.panichi@uslnordovest.toscana.it (V.P.); 5Department of Medical Sciences, University of Torino, 10126 Torino, Italy; federico.figliolini@gmail.com (F.F.); giovanni.camussi@unito.it (G.C.); 6Center for Autoimmune and Allergic Diseases (CAAD), University of Piemonte Orientale (UPO), Corso Trieste 15, 28100 Novara, Italy; 7Nephrology and Kidney Transplantation Unit, “Maggiore della Carità” University Hospital, Corso Mazzini 18, 28100 Novara, Italy

**Keywords:** endothelial progenitor cells, extracellular vesicles, glomerular endothelial cells, podocytes, glomerulonephritis, inflammation, angiogenesis, cytokines, complement cascade

## Abstract

Glomerulonephritis are renal inflammatory processes characterized by increased permeability of the Glomerular Filtration Barrier (GFB) with consequent hematuria and proteinuria. Glomerular endothelial cells (GEC) and podocytes are part of the GFB and contribute to the maintenance of its structural and functional integrity through the release of paracrine mediators. Activation of the complement cascade and pro-inflammatory cytokines (CK) such as Tumor Necrosis Factor α (TNF-α) and Interleukin-6 (IL-6) can alter GFB function, causing acute glomerular injury and progression toward chronic kidney disease. Endothelial Progenitor Cells (EPC) are bone-marrow-derived hematopoietic stem cells circulating in peripheral blood and able to induce angiogenesis and to repair injured endothelium by releasing paracrine mediators including Extracellular Vesicles (EVs), microparticles involved in intercellular communication by transferring proteins, lipids, and genetic material (mRNA, microRNA, lncRNA) to target cells. We have previously demonstrated that EPC-derived EVs activate an angiogenic program in quiescent endothelial cells and renoprotection in different experimental models. The aim of the present study was to evaluate in vitro the protective effect of EPC-derived EVs on GECs and podocytes cultured in detrimental conditions with CKs (TNF-α/IL-6) and the complement protein C5a. EVs were internalized in both GECs and podocytes mainly through a L-selectin-based mechanism. In GECs, EVs enhanced the formation of capillary-like structures and cell migration by modulating gene expression and inducing the release of growth factors such as VEGF-A and HGF. In the presence of CKs, and C5a, EPC-derived EVs protected GECs from apoptosis by decreasing oxidative stress and prevented leukocyte adhesion by inhibiting the expression of adhesion molecules (ICAM-1, VCAM-1, E-selectin). On podocytes, EVs inhibited apoptosis and prevented nephrin shedding induced by CKs and C5a. In a co-culture model of GECs/podocytes that mimicked GFB, EPC-derived EVs protected cell function and permeselectivity from inflammatory-mediated damage. Moreover, RNase pre-treatment of EVs abrogated their protective effects, suggesting the crucial role of RNA transfer from EVs to damaged glomerular cells. In conclusion, EPC-derived EVs preserved GFB integrity from complement- and cytokine-induced damage, suggesting their potential role as therapeutic agents for drug-resistant glomerulonephritis.

## 1. Introduction

The glomerulus is a crew of capillaries implicated in the ultrafiltration processes of the kidney. The glomerular capillary wall is composed of three layers: a fenestrated endothelium of glomerular endothelial cells, a glycocalyx with a complex mesh of proteins called glomerular basement membrane (GBM), and a layer of specialized visceral epithelial cells called podocytes [1]. Glomerular Filtration Barrier (GFB) has a very high hydraulic permeability combined with a marked selective permeability that excludes macromolecules such as albumin. Therefore, GFB retains most of the plasma proteins, with only 0.06% of albumin getting across the GBM [2]. In particular, podocytes constitute the slit diaphragms between their inter-digitating foot processes that prevent large molecules from reaching the urinary space [3]. The expression of nephrin in the podocyte slit diaphragm is crucial for maintaining GFB selectivity [4]. Injury to any of these three components can result in the development of proteinuria. In addition to external factors, several paracrine mediators released by resident glomerular or immune cells strictly regulate GFB integrity in precise cellular crosstalk [5,6]. In particular, the glomerular microenvironment maintains GEC function stimulating expression of endothelial receptors such as Platelet Endothelial Cell Adhesion Molecule-1 (PECAM-1) and Vascular Endothelial Growth Factor Receptor-2 (VEGFR-2) [7]. On podocytes, this crosstalk preserves cell function by maintaining the expression of nephrin [8,9]. Following GEC damage, different growth factors stimulate the migration of surviving cells to the injured site to repair vessels triggering angiogenesis [10,11].

Glomerulonephritis are inflammatory diseases affecting renal glomeruli able to compromise their filtering capacity and leading to chronic renal failure due to progressive fibrotic damage [12]. Activation of the complement cascade is a key factor for glomerulonephritis development and progression [13]. Complement protein fragment C5a induces the synthesis of pro-inflammatory cytokines (CKs) such as Interleukin-6 (IL-6) and Tumor Necrosis Factor (TNF-α) in the kidney, thus amplifying tissue damage [14,15]. TNF-α also increases the production of reactive oxygen species (ROS) [16] that produce multiple biological effects on glomeruli, including apoptosis or programmed cell death [17]; TNF-α induces in GECs an inflammatory phenotype by increasing interleukin-6 (IL-6) release and membrane expression of adhesion molecules such as ICAM-1, VCAM-1 and E-selectin [18,19]: these biological changes increase vascular permeability and leukocyte migration [17,20,21]. Moreover, TNF-α induces cell injury and loss of nephrin expression on podocytes disrupting the glomerular slit diaphragm [9,22,23]. IL-6 has an equally important role in glomerular cells, increasing inflammation and the recruitment of leukocytes [24].

Several studies have suggested that bone marrow-derived stem cells can repair injured glomeruli in experimental glomerulonephritis models [25,26]. In this context, endothelial progenitor cells (EPCs) are adult stem cells circulating in the peripheral blood to localize within sites of endothelial injury, triggering a regenerative program [27]. EPCs express both stem cell (CD34, CD133) and endothelial (VEGFR2, CD31) markers but they do not express monocyte (CD14) and platelet (P-selectin, CD41, CD42b) proteins [28,29]. Adhesion, tethering and rolling process of EPCs on endothelial cells is mediated by surface molecules such as L-selectin and various integrins [28,29].

At the level of the renal glomeruli, the main L-selectin ligands are the members of the CD34-superfamily, such as CD34 and podocalyxin [30]. In addition, the modification of the glycidic residues, such as the fucosylation of these molecules potentially induced by different injurious stimuli may allow the binding to L-selectin [28].

Injection of EPCs in experimental models of glomerulonephritis in rats with IgA nephropathy lowered disease progression by down-regulating the expression of inflammatory factors [31]; moreover, intra-renal injection of EPCs in the experimental rat model of Thy1.1 glomerulonephritis demonstrated a significant reduction of endothelial injury and complement-mediated mesangial cell activation [32].

The regenerative effect of EPCs is mainly ascribed to their ability to release paracrine mediators such as growth factors and extracellular vesicles (EVs) [33,34]. EVs have a critical role in intercellular communication by transferring proteins, lipids, and genetic information: EVs include different families such as exosomes and shedding vesicles that differ in size and intracellular formation [35,36]. We have previously demonstrated that EVs released from EPCs-activated angiogenesis in quiescent endothelial cells through the horizontal transfer of mRNAs [34]; moreover, we also observed that EVs protected the kidney from acute ischaemic injury by delivering pro-angiogenic and anti-apoptotic microRNAs [37]. Last, in the anti-Thy1.1 glomerulonephritis experimental model, we found EPC-derived EVs localized within injured glomeruli and inhibited complement-mediated mesangiolysis [38]. To demonstrate that the RNAs carried by the EVs mediated these effects, we have also investigated the biological effect of the EVs after treatment with the RNase, the enzyme able to degrade RNA [34,37,38].

In this study, we studied in vitro the protective effects of EPC-derived EVs on GECs and podocytes cultured with TNF-α, IL-6, and C5a, in an inflammatory microenvironment resembling that observed in glomerulonephritis.

## 2. Materials and Methods

### 2.1. Isolation and Characterization of Human EPCs

EPCs were isolated by density centrifugation from peripheral blood mononuclear cells (PBMC) of healthy donors, characterized and maintained in culture on fibronectin-coated plates as previously described [28].

### 2.2. Isolation and Characterization of Human EPC-Derived EVs

EVs were obtained from EPC supernatants by ultracentrifugation (Beckman Coulter Optima L-90K ultracentrifuge; Beckman Coulter, Fullerton, CA, USA) and characterized as previously described [34,37]. We resuspended EVs pellets in medium 199; we quantified protein content by the Bradford method (BioRad, Hercules, CA, USA), and we evaluated EV concentration, shape, and size by transmission electron microscopy and Nanosight analysis [37]: we stored EVs at −80 °C until use. In selected experiments, EVs were labelled with the red fluorescent dye PKH26 (Sigma Aldrich, St. Louis, MO, USA) or treated with 1 U/mL RNase (Ambion, Austin, TX, USA) [34].

#### 2.2.1. Nanoparticle Tracking Analysis

EPC-derived EV preparations were diluted (1:1000) in sterile 0.9% saline solution and analyzed by Nanosight LM10 (Nanosight, Amesbury, UK) equipped with the Nanoparticle Analysis System and NTA 1.4 Analytical Software. The Nanoparticle Tracking Analysis software allows the analysis of particle movement under Brownian motion in videos captured by Nanosight LM10 and calculates the diffusion coefficient, sphere equivalent, and hydrodynamic radius of particles by using the Strokes–Einstein equation. The concentration of EVs in supernatants was obtained by multiplying the instrument microparticles/mL value for the dilution. In addition, we analyzed all information on the size of the EVs in nm: mean, mode, median.

#### 2.2.2. RNA Extraction and Analysis

Total RNA from EPC-derived EVs was extracted using mirVana kit (Life Technologies, Carlsbad, CA, USA) and analyzed by NanoDrop1000 spectrophotometer. In addition, we assessed RNA quality by capillary electrophoresis on an Agilent 2100 Bioanalyzer (Agilent Technologies, Inc., Santa Clara, CA, USA) where the presence of total RNA (Agilent RNA 6000 Pico kit) and small RNAs (Agilent Small RNA kit) following manufacturer’s protocol.

#### 2.2.3. Guava FACS Analysis

FACS analysis was performed by the Guava easyCyte Flow Cytometer (Millipore, Billerica, MA, United States) and analyzed with InCyte software using the following FITC-, PE- or APC- conjugated antibodies: α4-integrin, α6-integrin (Miltenyi Biotec, Bergisch Gladbach, Germany), β1-integrin, L-selectin (BD Biosciences), αVβ3-integrin (Biolegend, San Diego, CA, USA). FITC, PE or APC mouse isotypic IgG (Miltenyi Biotec) were used as negative controls. Briefly, EPC-derived EVs (5 × 10^8^ particles) were resuspended in 100 µL of 0.1 µm filtered saline solution and incubated with antibodies for 15 min at 4°C; then samples were diluted in 200 µL filtered saline solution and acquired by the instrument.

### 2.3. Isolation and Characterization of Human Renal Glomerular Cells

Primary cultures of human glomerular endothelial cells (GECs) and podocytes were isolated from glomeruli from the cortical segment of kidneys of patients undergoing surgery for renal carcinomas. Cells were characterized and immortalized to obtain cell lines as previously described [39,40]. We cultured GEC lines in vitro on gelatin-coated flasks on EBM medium containing endothelial growth factors (Lonza, Basel, Switzerland) and podocyte cell lines in DMEM (GIBCO). All mediums contained 10% Fetal Bovine Serum (FBS, Hyclone, Logan, UT, USA) and 2 mM glutamine (GIBCO): for experimental procedures, we plated all cell lines in multi-well plates (Falcon Labware, Oxnard, CA, USA) [41]. In selected experiments, we incubated cells in an appropriate medium containing 20 ng/mL tumor necrosis factor (TNF)-α (Sigma Aldrich), 2.5 ng/mL IL-6, and 50 ng/mL human recombinant C5a protein (R&D Systems, Minneapolis, MN, USA) in the presence or absence of different concentrations of EPC-derived EVs assessed by Nanosight analysis.

### 2.4. Internalization of EPC-Derived EVs in GECs and Podocytes

We cultured GECs and podocytes on six-well plates or chamber slides (Thermo Scientific, Waltham, MA, USA). We incubated cells with PKH26-labelled EVs for 1 h, and then cells seeded on chamber slides were fixed with paraformaldehyde (Sigma Aldrich), nuclei were counterstained in blue by 2.5 μg/mL Hoechst (Sigma Aldrich), evaluated by confocal microscopy (Zeiss LSM 5 PASCAL, Jena, Germany). Cells cultured on six-well plates were detached by EDTA (Sigma) and analyzed by FACS (FACS Calibur, Becton Dickinson, Franklin Lakes, NJ, USA). In selected experiments, PKH26-labelled EVs were pre-incubated with 1 μg/mL of different antibodies directed to block the binding to αVβ-3 integrin (Biolegend, San Diego, CA, USA), α4-integrin, α6-integrin (Chemicon, Temecula, CA), CD29 or L-selectin (Becton Dickinson).

### 2.5. In Vitro Studies on Human GECs and Podocytes

#### 2.5.1. Angiogenesis

We studied the formation of capillary-like structures of GECs cultivated overnight on growth-factor reduced Matrigel (Becton Dickinson) on 24-well plates (5 × 10^4^ for well). We observed GECs under an inverted microscope at ×100 magnification (Leica DM IRE2 HC, Leica Microsystem, Deerfield, IL, USA).

#### 2.5.2. Proliferation

The 5 × 10^3^ GECs for the well were cultured on 96-well plates and incubated for 24 h with appropriate stimuli. GECs were then incubated for 24 h with 10 μM BrdU (Roche Diagnostics, Mannheim, Germany) and then analyzed in an automatized spectrophotometer at a wavelength of 405 nm, following the protocol of the manufacturer.

#### 2.5.3. Migration

We studied GEC migration under an inverted microscope. We calculated the net migratory speed using the MicroImage software (Casti Imaging, Venice, Italy) based on the straight-line distance between the starting and ending points divided by the time of observation.

#### 2.5.4. Gene Array Analysis

We used the Human GEarray kit to study angiogenesis on GECs (SuperArray Inc., Bethesda, MD, USA) to characterize the gene expression profile of cells cultured in the presence or absence of EVs. Microarray data archive: E-MEXP-3762, European Bioinformatics Institute: https://www.ebi.ac.uk/arrayexpress/experiments/E-MEXP-3762/, accessed on 30 November 2020).

#### 2.5.5. Prediction of miRNAs- Target Genes Interaction

We used miRNet (https://www.mirnet.ca/, accessed on 13 December 2020), a bioinformatics software that gives information about miRNA-target interactions and displays the association in a visual network [42]. We predict miRNAs involved in the down-regulation of the 16 genes identified by gene array analysis after searching in the miRNet human kidney database. We compared the suggested miRNAs by miRNet with previously identified miRNAs of EPC-derived EVs. (E-MEXP-2956, European Bioinformatics Institute: f2.5.8
www.ebi.ac.uk/arrayexpress/, accessed on 13 December 2020) [37].

#### 2.5.6. ELISA

We analyzed GEC supernatants for VEGF-A and HGF levels by ELISA (R&D Systems). We estimated their concentrations by generating a standard curve with appropriate controls according to the manufacturer.

#### 2.5.7. Immunofluorescence Studies

After appropriate stimuli for 24 h, GECs cultured in chamber slides were fixed with ethanol-acetic acid 2:1 and stained for 1 h at 4 °C with a polyclonal antibody directed to anti-VEGF or anti-CD31 (Santa Cruz Biotech, Santa Cruz, CA, USA). After extensive washing, GECs were incubated for 1 h at 4 °C with appropriate anti-isotype Alexa fluor-conjugated antibodies (Life Technologies, Carlsbad, CA, USA). We fixed cells with paraformaldehyde, performed nuclei counterstaining with 1 μg/mL propidium iodide (Sigma Aldrich), and analyzed samples on fluorescence microscopy ×400 magnification (Leica DM LA, Leica Microsystem). We assessed fluorescence intensity in 10 different microscopic fields for each experimental point by the ImageJ program (NIH, Bethesda, MD, USA).

In experiments with podocytes, after appropriate stimuli, we fixed cells in 4% paraformaldehyde for 15 min at 4 °C and incubated for 1 h at 4 °C with polyclonal antibody GP-N1 (Progen Biotechnik GmbH, Heidelberg, DE) to bind nephrin. After washing, we performed incubation for 40 min at 4 °C with Alexa Fluor-conjugated (Life Technologies) anti-guinea pig secondary antibodies. Finally, we performed nuclei counterstaining with 1 μg/mL propidium iodide (Sigma Aldrich), and we proceed to analyze samples on fluorescence microscopy at ×400 magnification (Leica DM LA, Leica Microsystem).

#### 2.5.8. PMN and PBMC Adhesion

After 12 h of stimulation in 24-well, we incubated GECs for 1 h with 5 × 10^4^/well polymorphonuclear neutrophils (PMNs) or PBMCs isolated from healthy volunteers and labelled with a 10 μm Vybrant cell tracer (Life Technologies). We fixed cells with paraformaldehyde, performed nuclei counterstaining with 1 μg/mL propidium iodide, and analyzed samples on fluorescence microscopy at ×400 magnification (Leica DM LA, Leica Microsystem). Samples were analyzed under a fluorescence microscope, counting green-stained cells in 10 different microscopic fields at ×200 magnification for each experimental point.

#### 2.5.9. FACS Analysis

We seeded GECs on six-well plates, and after appropriate stimuli for 24 h, cells were detached by EDTA and stained for 30 min at 4 °C with FITC- or PE-conjugated antibodies directed to bind ICAM-1, VCAM-1, E-Selectin (Beckton Dickinson). We used appropriate FITC- or PE-conjugated isotype antibodies as a negative control; FACS analysis was performed after fixation with paraformaldehyde 4% for 15 min at 4 °C. After appropriate stimuli, we detached cells by EDTA solution in experiments with podocytes, and we fixed cells with 4% paraformaldehyde solution for 15 min at 4 °C. We incubated cells with polyclonal antibody GP-N1, and after washing, we stained cells with FITC- (Sigma Aldrich) anti-guinea pig secondary antibodies incubation for 40 min at 4 °C before proceeding to FACS analysis.

#### 2.5.10. Cytotoxicity Assay

The 5 × 10^4^ GECs or podocytes were cultured on 24-wells and incubated for 24 h in different experimental conditions. At the end of this period, we incubated cells with XTT (Trevigen, Gaithersburg, MD, USA) in a medium lacking phenol red. After 1 h, we analyzed samples in an automatized spectrophotometer at a wavelength of 450 nm.

#### 2.5.11. Apoptosis

2 × 10^4^ GECs or podocytes were cultured on 96-well plates, incubated for 24 h with different stimuli, and then subjected to TUNEL assay following the manufacturer’s instructions (Apop-Tag; Oncor, Gaithersburg, MD, USA). Samples were analyzed under a fluorescence microscope, counting green-stained apoptotic cells in 10 different microscopic fields at ×100 magnification for each experimental point.

#### 2.5.12. Reactive Oxygen Species (ROS) Detection Assay

After 12 h of stimulation, we added 5-(and-6)-carboxy-2′,7′-dichlorodihydrofluorescein diacetate (carboxy-H2DCFDA) to GECs following the instructions of the manufacture (Image-iT LIVE Green ROS Detection Kit, Life Technologies); after 30 min cells were analyzed by FACS and immunofluorescence studies on confocal microscopy, as previously reported [43].

#### 2.5.13. Co-Culture of GECs and Podocytes

We seeded GECs on 24-well plates and stimulated them for 24 h. Then we changed the medium and put them on collagen-coated transwells with podocyte monolayers for 24 h (Corning Costar Corp., Cambridge, MA, USA). After stimulation, we put transwells in new plates, and we measured cytotoxicity, cell polarity, and permeability to albumin. For cytotoxicity, we used 250 μg/mL XTT (Sigma Aldrich) solution. Supernatants and filtrates were collected after 2 hr. and analyzed at a wavelength of 450 nm. Cell polarity was analyzed by measuring trans-epithelial electrical resistance (TEER) with an epithelial volt-ohm meter (EVOM, World Precision Instruments, Inc., Sarasota, FL, USA). We also evaluated permeability to albumin by diffusion of Trypan blue-albumin complexes across transwells. Aliquots of the medium from the upper and the lower wells were transferred to a 96-well plate and analyzed at the 590 nm wavelength (Model 680 Spectrophotometer, Biorad, Hercules, CA, USA). Results are expressed as arbitrary units (upper medium O.D./lower medium O.D.).

### 2.6. Statistical Analysis

We express all data of different experimental procedures as average ± 1 SD. We performed statistical analysis with ANOVA by Newmann–Keuls multi comparison test, and Student’s *t*-test when indicated. For FACS data, we performed the Kolmogorov Smirnov nonparametric statistical test. The significance level for all tests was set at *p* < 0.05. Data were analyzed using the GraphPad Prism 8.0.2 software. Data are expressed as mean ± 1 SD.

## 3. Results

### 3.1. Internalization of EPC-Derived EVs in Human Glomerular Cells

As shown by confocal microscopy studies (Figure 1A), we observed that PKH26 red-labeled EVs were efficiently internalized in vitro in GECs as well as in podocytes. FACS analysis showed that EVs stained both glomerular cell lines in a dose-dependent manner (Figure 1B). These results confirmed our previous in vivo findings on EV cell internalization in experimental Thy1.1 glomerulonephritis [38]. In selected experiments, we pre-incubated EVs with specific blocking antibodies (Ab), observing that Ab directed to L-selectin significantly inhibited EV internalization in both cell lines by approximately 50% (GECs positive control: 78.3 ± 3.3 %, GECs L-selectin: 39.4 ± 1.5 %; podocytes positive control: 58.9 ± 1.8 %, podocytes L-selectin: 25.8 ± 2.2 %; Figure 1C). By contrast, Abs directed to α4, α6, β1, and αVβ3 integrins affected less significantly EV internalization in GECs (αVβ3-integrin 64.7 ± 1.5 %, β1-integrin 63.6 ± 1.7 %, α6-integrin 71.2 ± 1.4 %, α4-integrin 62.1 ± 1.7 %, Figure 1C) and podocytes (αVβ3-integrin 37.2 ± 3.3 %, β1-integrin 39.5 ± 2.5 %, α6-integrin 51.4 ± 2.1 %, α4-integrin 34.5 ± 1.5 %, Figure 1C). The characterization of GECs and podocytes confirmed the importance of L-selectin in the internalization process. Both cell lines expressed L-selectin ligands in vitro. In particular, GECs expressed CD34, podocalyxin and fucosylated residues recognized by UEA-I lectin, whereas podocytes only expressed podocalyxin (Figure A1, Appendix B).

### 3.2. Effects of RNase Pre-Treatment of EPC-Derived EVs on Internalization in GECs and Podocytes

We compared the biological effects of EVs pre-treated with RNase with untreated EVs. We first observed that RNase did not change EV concentration. However, there was a significant reduction in size among the largest particles (Figure 2A,B). Of note, pre-treatment of EVs with RNase caused total RNA degradation (Figure 2C,D). In contrast, RNAse did not affect protein expression of EPC-derived EVs: L-selectin (EV: 52.9 ± 6.1 %; EV RNase: 51.6 ± 6.7 %), α4 (EV: 21.1 ± 6.1 %; EV RNase: 20.4 ± 7.5 %), α6 (EV: 18.6 ± 2.4 %; EV RNase: 18.4 ± 4.5 %), β1 (EV: 15.9 ± 3.9 %; EV RNase: 16.6 ± 4.6 %), αVβ3 integrin (EV: 23.8 ± 5.2 %; EV RNase: 23.1 ± 4.5 %), are comparably expressed in untreated and RNase pre-treated EVs (Figure 2E). Treatment with RNase did not affect the internalization of EPC-derived EVs in both GECs (EV: 80.5 ± 2.4 %, EV RNase 81.3 ± 2.9 %, Figure 2F) and podocytes (EV: 57.8 ± 2.9 %, EV RNase 58.6 ± 4.4 %, Figure 2F).

### 3.3. EPC-Derived EVs Triggered GEC Angiogenesis

In comparison to vehicle alone, EPC-derived EVs significantly enhanced (four-fold increase) the formation of capillary-like structures on Matrigel-coated plates (EV: 46.5 ± 8.8 capillary-like structures/field; vehicle: 11.6 ± 4.2 capillary-like structures/field; Figure 3A,B), promoted proliferation (EV: 1.042 ± 0.061 O.D. intensity; vehicle: 0.501 ± 0.103 O.D. intensity; Figure 3C), and migration by about tripling the speed of GECs in vitro (3 h EV: 14.2 ± 4.1 mm/hour; 3 h vehicle: 6.5 ± 1.8 mm/hour, Figure 2D) comparable to the positive control (FBS: 51.3 ± 5.2 capillary-like structures/field, 1.181 ± 0.184 O.D. intensity, 17.3 ± 2.9 mm/hour at 3 h). Pre-treatment of EPC-derived EVs with RNase abrogated all these effects (EV RNase: 13.8 ± 3. capillary-like structures/field, 0.467 ± 0.081 O.D. intensity, 5.9 ± 1.9 mm/hour at 3 h, Figure 3).

As demonstrated by PCR array, EPC-derived EVs modulated the expression of different genes involved in GEC angiogenesis (Figure 4). In particular, EPC-derived EVs increased the expression of the following genes: ANGPT1, ANPEP, CDH5, COL18A1, CXCL10, CXCL9, EFNA3, ENG, EREG, FGFR3, FLT1, HAND2, HGF, ID1, IFNA1, IFNB1, IFNG, IGF1, IL1B, ITGB3, JAG1, KDR, LECT1, LEP, MMP9, NOTCH4, PDGFA, PECAM1, PF4, PGF, PLAU, TGF-β1. We also observed 16 genes involved in GEC angiogenesis down-regulated by EVs: ANGPTL3, BAI1, COL4A3, CXCL1, CXCL6, S1PR1, EPHB4, FGF1, FGF2, FIGF, HPSE, ITGAV, LAMA5, NRP2, TGF-β2, THBS1. The significance of this data at the PCR array was confirmed by qRT-PCR analysis (Figure A2, Appendix B). In particular, gene expression increased four-fold for FLT1, HAND2, HGF, IGF1, KDR, LEP, MMP9, in the presence of EVs. Instead, the genes that are reduced by three times are ANGPTL3, S1PR1, FIGF, and THBS1.

Next, we analyzed the potential interacting miRNAs with these inhibited genes through the miRNet bioinformatic platform. Among the different suggested miRNAs, we confirmed the presence of 16 miRNAs carried by EPC-derived EVs that we had previously-identified [36]: miR-137; miR-142-3p; miR-142-5p; miR-17-3p; miR-17-5p; miR-18a; miR-19a; miR-30a-3p; miR-30e-3p; miR-30a-5p; miR-30e-5p; miR-324-5p; miR-425-5p; miR-484; miR-650 (Figure 5A). These molecules interact with one or more target genes downregulated by EPC-derived EVs (Figure 5B).

Immunofluorescence studies (Figure 6A) and FACS analysis (Figure 6B) showed that EPC-derived EVs significantly up-regulated GEC expression of PECAM-1 and VEGF-A in comparison to the vehicle (EV PECAM-1 68.3 ± 6.4 arbitrary units, vehicle: PECAM-1 43.5 ± 8.2 arbitrary units; EV VEGF-A 84 ± 7 arbitrary units, vehicle VEGF-A 42 ± 3 arbitrary units); moreover, EPC-derived EVs induced a five-fold increase of VEGF-A (vehicle: 331 ± 81 pg/mL; EV: 1635 ± 219 pg/mL, Figure 6C) and HGF (vehicle: 121 ± 41 pg/mL; EV: 674 ± 163 pg/mL Figure 6D) release in GEC supernatants as detected by ELISA. These effects were abrogated after incubation of GECs with EPC-derived EVs pre-treated with RNase (EV RNase: PECAM-1 40.5 ± 10.7 arbitrary units, VEGF-A 41 ± 4 arbitrary units, VEGF-A 359 ± 70 pg/mL, HGF 138 ± 52 pg/mL, Figure 6).

### 3.4. EPC-Derived EVs Protect GECs and Podocytes from Complement- and Cytokine-Mediated Injury

We evaluated the optimal stimulus timing of GECs and podocytes with pro-inflammatory cytokines 20 ng/mL TNF -α, 2.5 ng/mL IL-6, plus 50 ng/mL human recombinant C5a protein (CK). We considered 24 h as the optimal timing as there was a highly significant difference between the cells treated with CK and those treated with vehicle alone (Figure A3A,B, Appendix B).

Stimulation with CK has a cytotoxic (CK: 0.22 ± 0.25 O.D. intensity, vehicle: 1.12 ± 0.16 O.D. intensity, Figure 7A), pro-apoptotic (CK: 36.1 ± 12.6 apoptotic cells/field, vehicle: 4.7 ± 3.2 apoptotic cells/field, Figure 7B) and reactive oxygen species (ROS) production stimulating effects (CK: 91.4 ± 3.6 % of ROS-positive cells, vehicle: 38.6 ± 6.1 % of ROS-positive cells, Figure 7C,D) in comparison to vehicle alone. Then, we evaluated the ideal dose for stimulating GECs and podocytes with EPC-derived EVs in vitro in detrimental culture condition (CK) for 24 h (Figure A3C,D, Appendix B). The concentration of 25 μg/mL (10^9^ microparticles/mL) effectively protected both GECs and podocytes from CK-mediated cell damage (Figure A3C,D, Appendix B).

Incubation of GECs with EPC-derived EVs significantly maintained viability (0.85 ± 0.25 O.D. intensity, Figure 7A), resistance to apoptosis (10 ± 4.5 apoptotic cells/field, Figure 7B), and inhibited ROS production in comparison to cells treated CKs (43.5 ± 10.1% of ROS-positive cells Figure 7C,D). Pre-treatment of EVs by RNase abrogated the protective effects in terms of viability (0.35 ± 0.15 O.D. intensity), resistance to apoptosis (32.5 ± 11.2 apoptotic cells/field, Figure 7B) and limitation of ROS production (86.8 ± 10.6% of ROS-positive cells Figure 7C,D).

In selected experiments, we evaluated leukocyte adhesion to GEC monolayer after incubation in different culture conditions. After 24 h, GEC monolayers were washed and incubated for 2 h with FITC-labelled PBMC or PMN and then samples were evaluated under a UV light microscope.

CK stimulation doubled the number of PBMCs (vehicle: 12.2 ± 2.0; CK: 22.2 ± 3.1 adherent PBMCs/field) and quadrupled the number of PMNs (vehicle: 7.6 ± 2.2; CK: 30.8 ± 10.9 adherent PMNs/field) adhering to GEC monolayers (Figure 8A,B).

EPC-derived EVs inhibited PBMC (10.6 ± 3.2 adherent PBMCs/field) and PMN (7.6 ± 2.2 adherent PMNs/field, Figure 8A,B) adhesion to GEC monolayers cultured in an inflammatory micro-environment.

At the same time, compared to the treatment with vehicle alone, we observed a 3-fold increase in the percentage of positive GECs for ICAM-1, VCAM-1, E-selectin after 24 h of treatment with CKs.

The addition of EPC-derived EVs significantly down-regulated ICAM-1, VCAM-1, and E-selectin expression in GECs (Figure 8C). Again, the pre-treatment of EVs with RNase had no biological effects on adhesion molecule expression (Figure 8C).

We observed similar results on podocytes: in comparison to vehicle alone, cells treated with pro-inflammatory C5a and CKs showed a significant decrease of cell viability (vehicle: 1.25 ± 0.11, CK:0.57 ± 0.13 O.D. intensity, Figure 9A), and resistance to apoptosis (vehicle: 3.4 ± 2.2, CK:28.5 ± 5.8 apoptotic cells/field, Figure 9B). When podocytes were treated with EPC-derived EVs, viability (1.10 ± 0.16 O.D. intensity, Figure 9A) and resistance to apoptosis (13.2 ± 2.1 apoptotic cells/field, Figure 9B) were maintained. However, RNase treatment of EPC-derived EVs did not preserve viability (0.52 ± 0.20 O.D. intensity, Figure 9A) and resistance to apoptosis of podocytes (30.3 ± 4.7 apoptotic cells/field Figure 9B).

We also evaluated the ability of EPC-derived EVs to protect podocytes from nephrin shedding induced by CK. For this purpose, we incubated podocytes for 24 h with EVs, RNAse pre-treated EVs or vehicle alone. At the end of incubation, we stimulated podocytes with CK for 1 h, and we observed a halving of nephrin expression on the cell surface (Figure 9C). EPC-derived EVs, but not RNase-pre-treated EVs, prevented nephrin shedding from podocytes (Figure 9).

### 3.5. Protective Role of EPC-Derived EVs in a GEC-Podocyte Co-Culture Model Mimicking GFB

We simulated an in vitro co-culture model of GFB by culturing GECs for 24 h under different experimental conditions. At the end of this stimulus, we left GECs in a new medium in contact with a transwell with a monolayer of podocytes inside. Over the next 24 h, we studied the effect of factors released by GECs on podocytes (Figure 10A).

The treatment of GECs with CKs induced on podocytes a significant decrease of viability (vehicle: 1.86 ± 0.61 O.D. intensity, CK: 1.03 ± 0.18 O.D. intensity, Figure 10B), induced functional alterations such as loss of cell polarity assessed by TEER (vehicle: 837 ± 234 ohm/cm^2^, CK: 97 ± 43 ohm/cm^2^, Figure 10C) and increased permeability to albumin (vehicle: 0.49 ± 0.19 arbitrary units, CK: 0.90 ± 0.26 arbitrary units, Figure 10D) in comparison to controls. These effects on podocytes were significantly decreased by treatment of GECs with EPC-derived EVs (CK+ EV: 2.01 ± 0.32 O.D. intensity, 774 ± 293 ohm/cm^2^, 0.53 ± 0.17 arbitrary units, Figure 10) but not with RNase-treated EVs (CK+ EV RNase: 1.05 ± 0.29 O.D. intensity, 109 ± 58 ohm/cm^2^, 0.85 ± 0.31 arbitrary units, Figure 10). These results suggest a potential positive effect of EPC-derived EVs in the mechanisms of GEC-podocyte interaction in a GFB model.

## 4. Discussion

In the course of glomerulonephritis, inflammatory cytokines and activation of the complement cascade are key factors for the development of acute alterations of typical glomerular architecture and function and for the progression toward glomerulosclerosis and chronic kidney disease (CKD) [44,45,46,47,48,49]. Rarefaction of glomerular capillaries consequent to endothelial injury and alterations of podocyte functions, such as the loss of expression of the slit diaphragm protein, nephrin has been shown to contribute to these detrimental biological processes [50,51].

Recently, extracellular vesicles (EVs) have been described as paracrine mediators released by resident glomerular cells [5]. EVs are a heterogeneous population of micro-organoid bodies that include exosomes and microvesicles that present different size, antigenic composition, and functional properties for the cargo of proteins and RNA subsets [35]. Of interest, stem cell-derived EVs have the property of repairing damaged tissues [52]: indeed, the regenerative properties of stem cells are mediated by the release of paracrine factors such as growth factors and EVs, rather than by replacing cells lost after tissue injury.

EPCs are bone marrow-derived stem cells circulating in the peripheral blood that can localize within endothelial injury sites [27]. EPCs can trigger a regenerative program by revascularizing damaged tissues favoring angiogenesis by secreting growth factors and other paracrine mediators, such as EVs [33,53,54].

The first aim of this study was to evaluate the biological activities of EPC-derived EVs on human renal glomerular cells in vitro. We observed that EPC-derived EVs internalized efficiently in human GECs, and podocytes through a mechanism mainly mediated by L-selectin, confirming the data previously observed in other experimental models [38]: indeed, antibodies directed to several integrins (α4, α6, β1, and αVβ3 inhibited less efficiently the internalization of EPC-derived EVs. These results suggest that L-selectin is the key molecule expressed on the EPC surface essential for homing on sites of vascular injury [28] and for the internalization of EPC-derived EV in human renal glomerular cells. Furthermore, we herein demonstrated that both GECs and podocytes express L-selectin ligands in vitro: podocytes express podocalyxin [30], and GECs also express CD34 and fucosylated residues recognized by UEA-I [28].

Since we have previously demonstrated that EVs released from EPCs triggered an angiogenic program in quiescent endothelial cells by a horizontal transfer of mRNA [34] and protected the kidney from acute ischemic injury by delivering their RNA content inducing hypoxic-resident renal cells to a regenerative program [37], the second purpose of the present study was to investigate whether EPC-derived EVs induced specific biological effects also on human GECs. We observed that EVs triggered angiogenesis in GECs by increasing the formation of capillary-like structures, proliferation, and migration in vitro. These effects were abrogated by pre-treatment of EVs with RNase, suggesting that horizontal RNA transfer is fundamental for EV-induced biological activity. These data confirmed previous findings found on endothelial cells of different tissue origin [34,37,55]. We also investigated whether RNase treatment only degraded RNA or had other effects on EV function. We found that RNase treatment degraded all RNA subsets within EVs, but not surface proteins without affecting EV internalization in target glomerular cells. These results further suggest the relevance of RNA transfer from EPC EVs to injured glomerular cells to induce their regenerative program.

The relevance of angiogenesis in the glomerular filtration barrier damage in the course of inflammatory diseases such as glomerulonephritis has been demonstrated in different experimental models [56,57]: moreover, a potential role of angiogenesis has been recently described also in human glomerular diseases [58,59]. Several mediators, including Vascular Endothelial Growth Factor (VEGF-A) and nitric oxide (NO) have been shown to play a pivotal role in glomerular capillary repair during inflammatory diseases [60,61]. Furthermore, the expression of angiopoietins in the glomerulus suggested a mechanism for the maintenance of the glomerular endothelium and modulation of the actions of glomerular VEGF in inflammatory diseases as well as in the recovery from them [62]. The lack of the matricellular protein thrombospondin-2 (TSP-2) in mice is known to accelerate renal injury: TSP2 is a major endogenous antiangiogenic and matrix metalloproteinase 2-regulating factor in renal diseases [63]. Last, the inhibition of another anti-angiogenic factor such as Endoglin has been shown to promote intussusceptive angiogenesis in experimental nephritis [64]. Of interest, the administration of bone marrow-derived angiogenic cells has been previously shown to reduce endothelial injury and mesangial activation in anti-Thy1.1 glomerulonephritis: the incorporation into the glomerular endothelial lining and production of angiogenic factors contributed to these protective effects [32]. Another study showed that bone marrow-derived progenitor cells participate in glomerular endothelial and mesangial cell turnover and contribute to microvascular repair [65]. For all these reasons, we investigated the angiogenic pathways induced by EPC-derived EVs in GECs. EVs increased mRNA expression of endothelial cell receptors endoglin (ENG/CD105), platelet/endothelial cell adhesion molecule (PECAM-1), and vascular endothelial growth factor receptor type 2 (VEGFR-2): these data were confirmed at the protein level. Furthermore, EVs up-regulated adhesion molecules involved in endothelium integrity such as CDH5, but down-regulated ITGAV and integrin β3 gene expression. We also observed an increased expression of pro-angiogenetic genes involved in glomerular cell crosstalk [5,6]: angiopoietin-1, FLT1, HGF, PDGF-α, TGF-β1. To confirm intraglomerular pathways modulated by EPC-derived EVs, we found that GECs released high levels of HGF and VEGF-A in supernatants. EVs also increased other growth factors such as IGF1, PGF, EREG, but decreased FGF-1 and FGF-2 and FIGF gene expression. Moreover, we found the up-regulation of genes of trans-membrane receptors such as NOTCH4, FGFR3, and the down-regulation of NRP2, EPHB4, S1PR1. LEP, a molecule associated with endothelial cell differentiation during angiogenesis [14], was increased in GECs after EV treatment. EPC-derived EVs also decreased anti-angiogenic genes such as THBS1, BAI1, and pro-fibrotic genes [66], including TGF-β2. EVs modulated the gene expression of glomerular basement membrane proteins by up-regulating MMP9, COL4A3, and down-regulating LAMA5 and COL18A1. Moreover, the expression of other exoenzymes involved in angiogenesis was modulated by EVs (PLAU, ANPEP, and HPSE).

Interestingly, EVs inhibited GEC expression of angiopoietin-like 3, a molecule that increases endothelial cell barrier permeability in glomeruli [10] and PF4, a negative regulator of mesangial cell proliferation [67], thus suggesting other further potential protective mechanisms on GFB integrity. Moreover, we found in GECs an increased mRNA expression of EFNA3, ID1, and JAG1, all genes involved in proliferation and migration of endothelial cells [68] as well as EPCs [69,70]. The most EV-induced gene was HAND2, a molecule that belongs to the Twist family, involved in the development of different organs [71]: however, its biological function in kidney glomerulus is still unknown. Another inexplicable up-regulated gene by EPC-derived EVs was LECT1, a protein known to promote chondrocyte growth, inhibit angiogenesis, but with an unknown function in the kidney.

In previous studies, we identified more than 150 miRNAs carried by EPC-derived EVs [37]. We have herein identified 16 microRNAs among those carried by EVs able to down-regulate anti-angiogenic genes. In fact, 12 of these miRNAs interact with THBS1, an important negative regulator of angiogenesis. Interestingly, miR-17-5p is potentially able to bind other 6 mRNAs in addition to THBS1: COL4A3, EPHB4, ITGAV, NRP2, S1PR1, TGF-β2. In addition, 12 further miRNAs can potentially interact with two or more target mRNAs: on this basis, we could speculate that EV-induced GEC angiogenesis is mediated by the concomitant action of different miRNAs able to interact with several mRNAs within target cells.

Another relevant aim of this study was to evaluate whether EPC-derived EVs could protect renal glomerular cells cultured in detrimental inflammatory conditions. For this purpose, we decided to re-create an inflammatory microenvironment (CK) by using a mix of cytokines (TNF-α, IL-6) and the complement fraction C5a on glomerular cells in vitro and to study the effect of EPC-derived EVs in this setting. CKs induced activation in GECs of a pro-inflammatory phenotype, increasing the expression of adhesion molecules such as ICAM-1, VCAM-1, and E-selectin [17,20,21] and leading to the enhanced recruitment of PBMCs and PMNs. All these events are mediated by a reactive oxygen species (ROS) [16] that, in the presence of prolonged inflammatory stimuli, increase GEC cytotoxicity and death by apoptosis [16].

On podocytes, CKs induced cell injury and triggered apoptosis like on GECs, but the early observed effects are the rapid loss of nephrin expression by shedding [9,22,23]. Nephrin is a protein located at the podocyte slit diaphragm essential for the preservation of glomerular permeselectivity. Nephrin is down-regulated by drugs (REF mTOR Biancone) and by inflammatory mediators leading to the development and worsening of proteinuria [72].

EPC-derived EV inhibited all these effects of GECs and podocytes treated with CK. By acting on different mechanisms, such as inflammation or the activation of regenerative mechanisms on glomerular cells, EVs are one of the potential treatments for redoubtable and unpredictable diseases such as glomerulonephritis.

We have previously described that EVs derived from EPC exert a protective effect in Thy1.1 glomerulonephritis on rats by inhibiting antibody- and complement-mediated injury of mesangial cells [38]. EVs protected glomeruli by fibrotic processes that lead inexorably to chronic renal failure. Moreover, in the same experimental model, EV treatment preserved endothelial- (RECA-1) and podocyte-markers (synaptopodin) expression suggesting a role of EVs in protecting these glomerular cells. In the light of the data herein described, we could speculate that EPC-derived EVs can not only antagonize complement cascade, but also inflammatory injury mediated by CKs.

On this basis, we set up a co-culture model of GECs and podocytes to mimic GFB in vitro and to evaluate the potential crosstalk between the two cell types. Of interest, we observed that pre-stimulation of GECs with EPC-derived EVs maintained podocyte viability, trans-epithelial electrical resistance, and inhibited loss of permeability to albumin, all established markers of GFB integrity. All these effects on podocytes might be related to the pre-angiogenic phenotype induced by EPC-derived EV on GECs and mediated by the release of growth factors such as HGF and VEGF-A.

We also confirmed in this GEC-podocyte co-culture model that RNase pre-treatment abrogated all the effects mediated by EPC-derived EVs, suggesting that mRNA and microRNA transfer plays a critical role in these biological effects as previously described in other experimental models [34,37,38,55].

## 5. Conclusions

EPC-derived EVs may preserve GFB integrity from complement- and cytokine-induced damage: this protective effect on glomerular cells seems to be mainly ascribed to RNA transfer from progenitor-derived EVs to injured GECs and podocytes. Based on previously-published data in experimental glomerulonephritis models and on the results of the present study, EPC-derived EV could represent an attractive alternative in patients resistant to classical therapeutic agents. Moreover, EVs can induce immunomodulation and glomerular healing without the potential adverse effects of stem cell therapy, including maldifferentiation and tumorigenesis.

## Figures and Tables

**Figure 1 cells-10-01675-f001:**
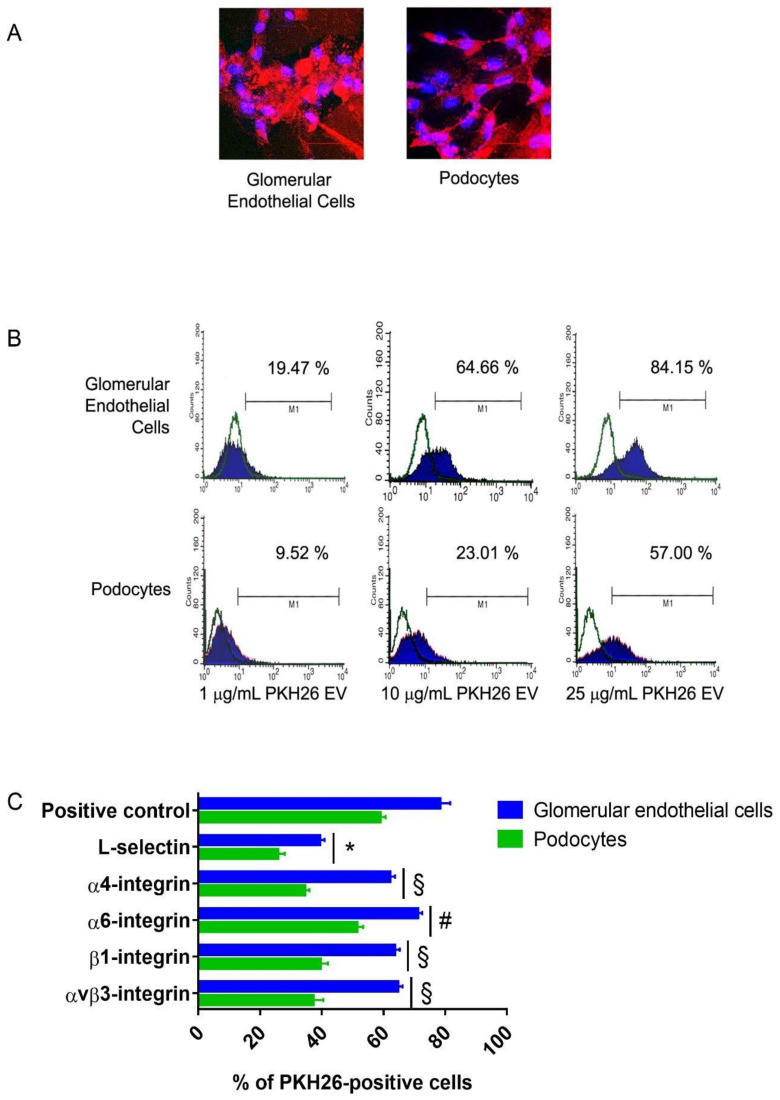
Internalization of EPC-derived EVs in human kidney glomerular cells in vitro. (**A**) Representative confocal microscopy micrographs showing the internalization of 25 μg/mL (25*10^9^ particles/mL) PKH26 red dye-labelled EPC derived EVs in human GECs and podocytes. Nuclei were counterstained in blue by 2.5 μg/mL Hoechst (magnification x400, scale bar 50 μm). (**B**) Representative FACS analysis of dose-response PKH26-labelled EV internalization in human GECs and podocytes. (**C**) graph showing FACS analysis of PKH26-labelled EV internalization in GECs and podocytes. We expressed results as the mean percentage of positive cells ± 1 SD. ANOVA performed statistical analysis with Newmann–Keuls’s multiple comparison test and the Kolmogorov–Smirnov test. Pre-incubation of all cell lines with PKH26-labelled EVs with 1 μg/mL blocking mAb directed to L-selectin significantly inhibited EV internalization compared to a positive control (* *p* < 0.05 L-selectin vs. Positive Control). The inhibition of internalization was not modulated when PKH26-labeled EVs were pre-incubated with blocking mAbs directed to α4, β1, αVβ3 integrin (§ *p* < 0.05 α4, β1 or αVβ3 vs. L-selectin; * *p* < 0.05 α4, β1 or αvβ3 vs. Positive Control), in particular with the mAb directed to α6 integrin (# *p* < 0.05 α6 vs. α4, β1 or αVβ3; § *p* < 0.05 α6 vs. L-selectin; * *p* < 0.05 α6 vs. Positive Control). We performed three experiments with similar results.

**Figure 2 cells-10-01675-f002:**
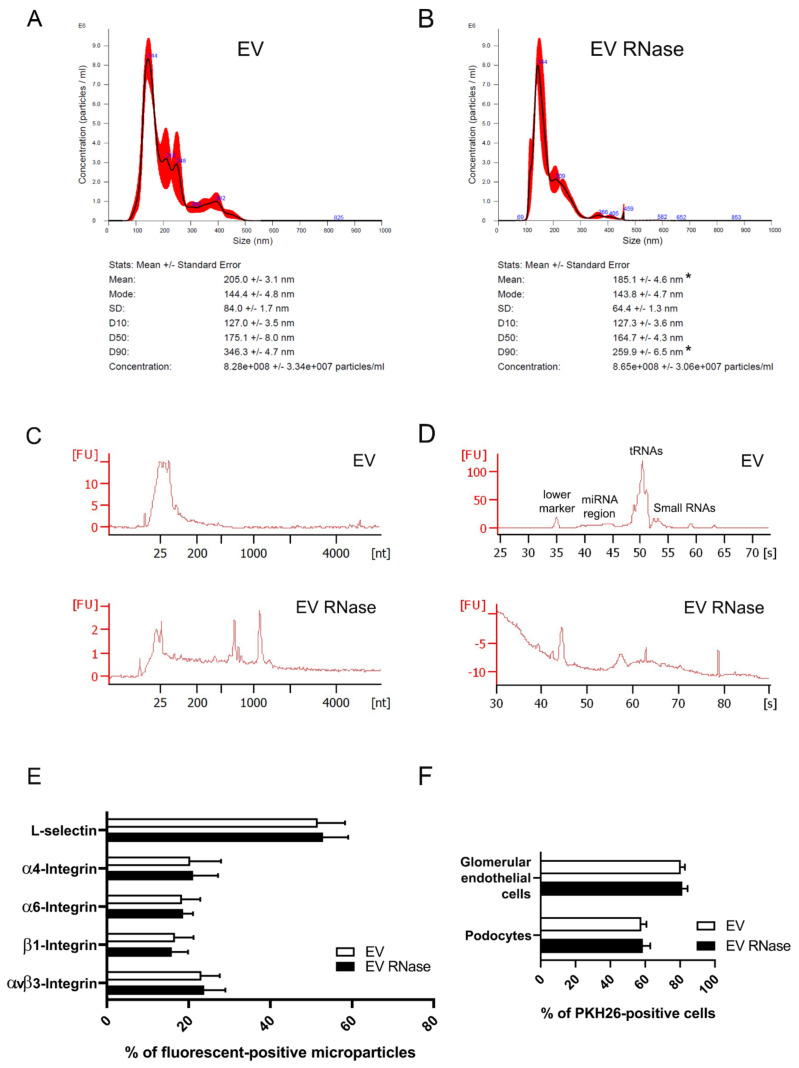
Characterization and internalization of EPC-derived EVs EVs pre-treated or not with 1 U/mL RNase. (**A,B**) Nanosight analysis of purified EVs (**A**) and EVs pre-treated with RNase (**B**); the open curve shows the relationship between particle number distributions (left Y-axis) and particle size (X-axis). The curve has a red border because it takes into account the analyzes of 3 samples. No statistical differences are observed after RNase treatment in terms of mode size (EV 144.4 ± 4.8 nm vs. EV RNase 143.8 ± 4.7 nm), and 10th percentile size (D10, EV 127.0 ± 3.5 nm vs. EV RNase 127.3 ± 3.6 nm), median size (D50, EV 175.1 ± 8.0 nm vs. EV RNase 164.7 ± 4.3 nm) particle concentration (EV 8.28 × 10^8^ ± 3.34 × 10^7^ particles/mL vs. EV RNase 8.65 × 10^8^ ± 3.06 × 10^7^ particles/mL). We observed statistical difference of mean size (* *p* < 0.05, EV 205.0 ± 3.1 nm vs. EV RNase 185.1 ± 4.6 nm) and 90th percentile size (D90, * *p* < 0.05, EV 346.3 ± 4.7 nm vs. EV RNase 259.9 ± 6.5 nm). Results are expressed as Average number ± 1 SD. (**C**,**D**) Representative bioanalyzer micrographs showing the total RNA (**C**) and small RNA (**D**) content of EVs treated with vehicle alone or with 1 U/mL RNase. In **C** we can see an RNA content mainly small RNAs in the EVs, peaked at 25 nucleotides in size, RNA Concentration (1.165 pg/µL) and RNA Integrity Number (RIN) is 2.6; in the sample of EVs pre-treated with RNAse, RNA is degraded as shown by the representative curve, the reduced concentration value on the y axis and RIN is undetermined. In (**D**), small RNAs profiling in EVs is reported: small RNA Concentration (6848,3 pg/µL), miRNA Concentration (2974,4 pg/µL), miRNA/Small RNA Ratio (43 %). In the sample of EVs pre-treated with 1 U/mL RNAse, RNA is degraded as shown by the representative curve and inability to calculate any value and index of integrity and concentration. (**E**) GUAVA FACS showing protein expression in EVs and EVs pre-treated with 1 U/mL RNase. There are no statistically significant differences between untreated (white columns) and RNase pre-treated EVs (black columns) in terms of L-selectin, α4, α6, β1, αVβ3 integrin expression. We expressed results as the mean of the percentage of fluorescent-positive microparticles ± 1 SD. (**F**) Internalization of untreated and 1 U/mL RNase pre-treated EVs human kidney glomerular cells. Graph showing FACS analysis of PKH26-labelled untreated (white columns) and PKH26-labelled 1 U/mL RNase pre-treated EV (black columns) internalization in GECs and podocytes. We expressed results as the mean of the percentage of positive cells ± 1 SD. We performed three experiments and the statistical analysis by ANOVA with Newmann–Keuls’s multiple comparison test and the Kolmogorov–Smirnov test.

**Figure 3 cells-10-01675-f003:**
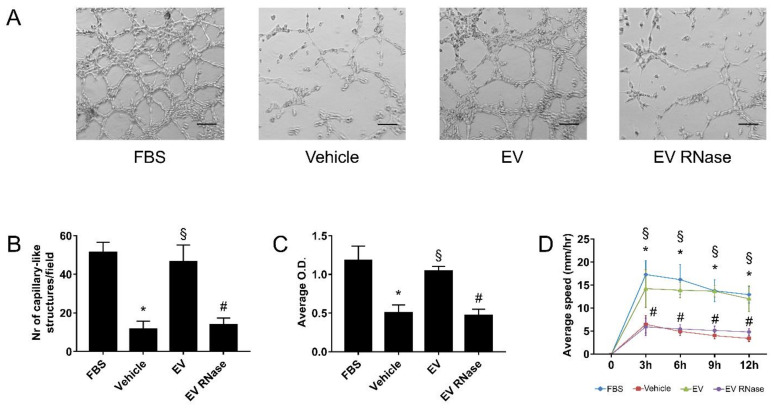
EPC-derived EVs induced GEC angiogenesis by stimulating proliferation and migration. (**A**,**B**) Representative light microscopy images (**A**) and count (**B**) of GECs cultured on Matrigel in vitro in different experimental conditions. Original magnification ×100; scale bar 50 μm. We expressed data as the average number of capillary-like structures/field ± 1 SD. (**C**,**D**) Analysis of GEC proliferation by BrdU assay (**C**) and migration test on six-well plates after 3, 6, 9, and 12 h (**D**). For the BrdU assay, we expressed data as average O.D. intensity ± 1 SD; for migration test, we express the data like average speed (mm/hour) ± 1 SD. We performed three experiments with similar results; we performed the statistical analysis by ANOVA with Newmann–Keuls multiple comparison tests and Student’s *t*-test. In comparison to standard culture conditions with fetal bovine serum (FBS), serum deprivation (vehicle) inhibited the formation of the number of capillary-like structures per field (**B**), proliferation (**C**), migration (**D**, * *p* < 0.05 vehicle vs. FBS). EPC-derived EVs significantly increased these effects (§ *p* < 0.05 EV vs. vehicle), that were abrogated by RNase pre-treatment of EVs (# *p* < 0.05 EV RNase vs. EV). We performed the statistical analysis by ANOVA with Newman–Keuls multiple comparison test and Student’s *t*-test.

**Figure 4 cells-10-01675-f004:**
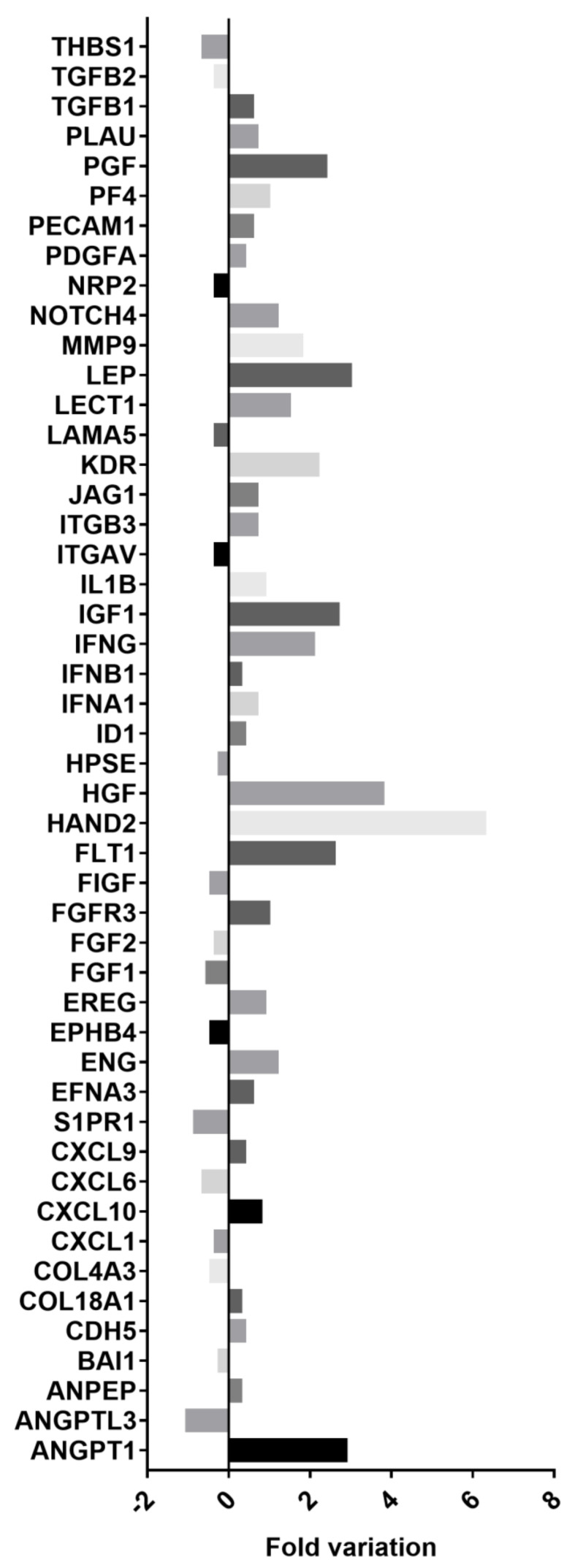
Intracellular pathways involved in glomerular endothelial cells (GEC) angiogenesis induced by Endothelial Progenitor Cells (EPC)-derived extracellular vesicles (EVs). RT-PCR array analysis of GEC incubated with EPC-derived EV vs. vehicle alone (angiogenesis-related genes). The graph shows the fold variation of expression of angiogenesis-related genes in GECs stimulated with EVs compared to GECs treated with vehicle alone. We normalized samples for the signals found in housekeeping genes (actin β, Hypoxanthine phosphoribosyltransferase 1, Ribosomal protein, large P0, GAPDH, β2-microglobulin). We performed three different experiments with similar results. Gene table: ANGPT1: angiopoietin 1; ANGPTL3: Angiopoietin-like 3; ANPEP: Alanyl (membrane) aminopeptidase; BAI1: Brain-specific angiogenesis inhibitor 1; CDH5: Cadherin 5, type 2 (vascular endothelium); COL18A1: Collagen, type XVIII, α1; COL4A3: Collagen, type IV, α3 (Goodpasture antigen); CXCL1: Chemokine (C-X-C motif) ligand 1 (melanoma growth stimulating activity, α); CXCL10: Chemokine (C-X-C motif) ligand 10; CXCL6: Chemokine (C-X-C motif) ligand 6; CXCL9: Chemokine (C–X–C motif) ligand 9; S1PR1: Sphingosine-1-phosphate receptor 1; EFNA3: Ephrin-A1; ENG: endoglin/CD105; EPHB4: EPH receptor B4; EREG: Epiregulin; FGF1: Fibroblast growth factor 1 (acidic); FGF2: Fibroblast growth factor 2 (basic); FGFR3: Fibroblast growth factor receptor 3; FIGF: C-fos induced growth factor (vascular endothelial growth factor D); FLT1: vascular endothelial growth factor type 1/vascular permeability factor receptor; HAND2: hearth and neural crest derivatives expressed; HGF: hepatocyte growth factor; HPSE: Heparanase; ID1: Inhibitor of DNA binding 1, dominant negative helix-loop-helix protein; IFNA1: Interferon, α1; IFNB1: Interferon, β1; IFNG: Interferon, γ; IGF1: Insulin-like growth factor 1 (somatomedin C); IL1B: Interleukin, 1β; ITGAV: Integrin, αV (vitronectin receptor, α polypeptide, antigen CD51); ITGB3: Integrin, β3 (platelet glycoprotein IIIa, antigen CD61); JAG1: Jagged 1; KDR: (FLK-1) vascular endothelial growth factor receptor type 2; LAMA5: Laminin Subunit α5; LECT1: Leukocyte cell derived chemotaxin 1; LEP: Leptin; MMP9: matrix-metal protease 9; NOTCH4; NRP2: Neuropilin 2; PDGFA: Platelet-derived growth factor α polypeptide; PECAM1: platelet/endothelial cell adhesion molecule (CD31 antigen); PF4: Platelet factor 4; PGF: placental growth factor; PLAU: Plasminogen activator, urokinase; TGFB1: Transforming growth factor, β1; TGFB2: Transforming growth factor, β2; THBS1: Thrombospondin 1.

**Figure 5 cells-10-01675-f005:**
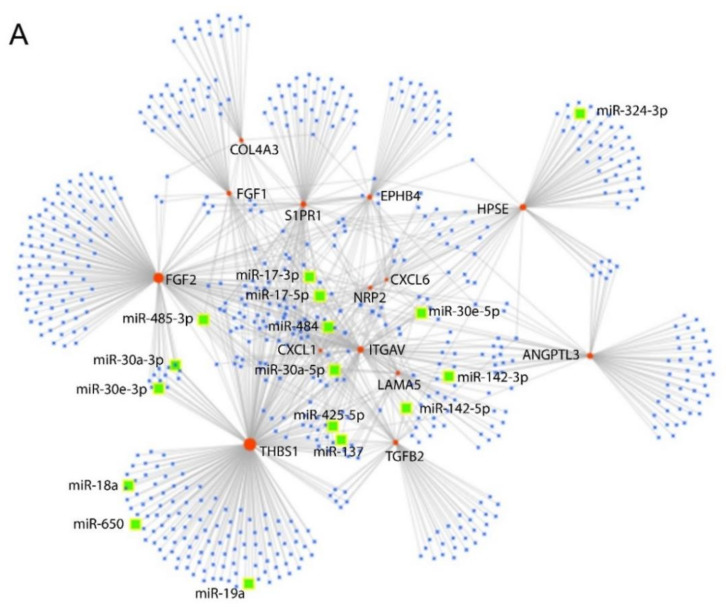
Network of predicted miRNAs-target genes interaction. (**A**) We observed 1038 potential interactions between the miRNAs (blue spots) predicted by miRNet to inhibit the expression of genes (orange spots). Between the miRNAs we observed that 16 of them, there were 16 miRNAs carried by EPC-derived EVs (green spots): miR-137; miR-142-3p; miR-142-5p; miR-17-3p; miR-17-5p; miR-18a; miR-19a; miR-30a-3p; miR-30e-3p; miR-30a-5p; miR-30e-5p; miR-324-3p; miR-425-5p; miR-484; miR-485-3p, miR-650. (**B**) Table showing the miRNAs present in EPC-derived EVs and their target genes. ANGPTL3: Angiopoietin-like 3; COL4A3: Collagen, type IV, α3 (Goodpasture antigen); CXCL1: Chemokine (C–X–C motif) ligand 1 (melanoma growth stimulating activity, α); CXCL6: Chemokine (C–X–C motif) ligand 6; S1PR1: Sphingosine-1-phosphate receptor 1; EPHB4: EPH receptor B4; FGF1: Fibroblast growth factor 1 (acidic); FGF2: Fibroblast growth factor 2 (basic); HPSE: Heparanase; ITGAV: Integrin, αV (vitronectin receptor, α polypeptide, antigen CD51); LAMA5: Laminin Subunit α5; NRP2: Neuropilin 2; TGFB2: Transforming growth factor, β2; THBS1: Thrombospondin 1.

**Figure 6 cells-10-01675-f006:**
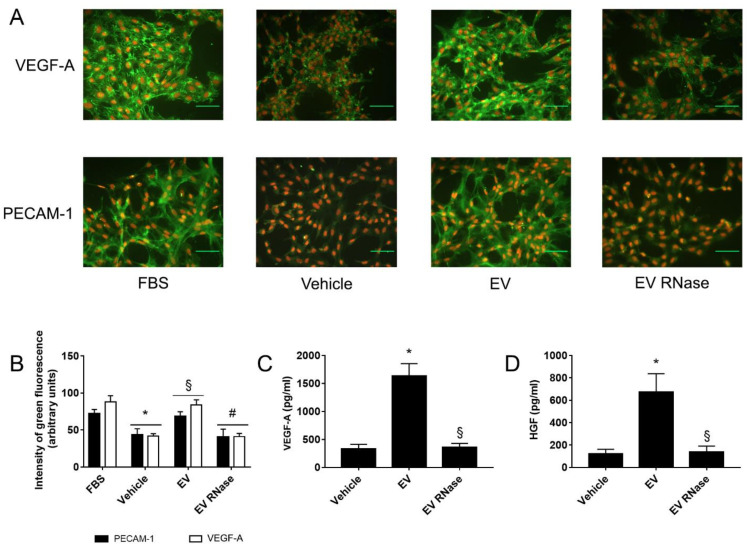
EPC-derived EVs induced GEC angiogenesis through the release of pro-angiogenic factors. (**A**,**B**) Representative immunofluorescence micrographs (**A**) and quantification on fluorescence intensity of platelet/endothelial cell adhesion molecule (PECAM-1) and vascular endothelial growth factor (VEGF-A) expression (**B**, green staining) in GECs. Nuclei were counterstained with 1 μg/mL propidium iodide; original magnification x100; scale bar: 50 μm. We expressed data as mean arbitrary units ± 1 SD of three different experiments for quantification on fluorescence intensity. We performed three experiments with similar results; we performed the statistical analysis by ANOVA with Newmann–Keuls multiple comparison tests and Student’s *t*-test. Compared to normal culture conditions with fetal bovine serum (FBS), serum deprivation (vehicle) inhibited PECAM-1 and VEGF-A expression. EPC-derived EVs significantly increased both protein expression (§ *p* < 0.05 EV vs. vehicle), which was abrogated by RNase pre-treatment of EVs (# *p* < 0.05 EV RNase vs. EV). (**C**,**D**) ELISA for VEGF-A (**C**) and HGF (**D**) on supernatants of GECs incubated with different culture conditions. We expressed results as mean pg/mL ± 1 SD of three different experiments. We performed statistical analysis was performed by ANOVA with Newman–Keuls multiple comparison test and Student’s *t*-test. EPC-derived EVs significantly increased the release of both growth factors by GECs (* *p* < 0.05 EV vs. vehicle). By contrast, RNase pre-treatment of EVs abrogated the release of VEGF-A and HGF (§ *p* < 0.05 EV RNase vs. EV).

**Figure 7 cells-10-01675-f007:**
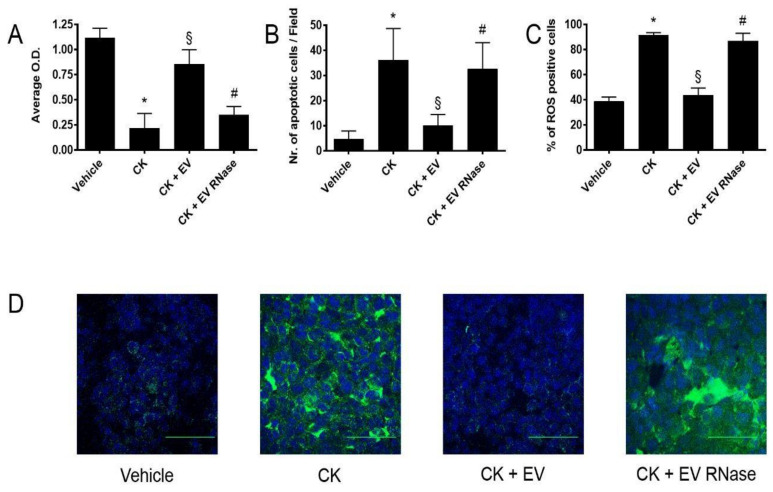
EPC-derived EVs protected GECs from complement- and cytokine-induced damage (**A**,**B**). Graphs showing GEC cytotoxicity by XTT assay (**A**) and apoptosis by TUNEL assay (**B**). For XTT assays, data are expressed as average OD intensity ± 1 SD, whereas we expressed TUNEL assays data as the average number of green fluorescent apoptotic cells ± 1 SD. We performed three experiments with similar results for all the assays and the statistical analysis by ANOVA with Newmann–Keuls multiple comparison test and Student’s *t*-test. (**C**,**D**) FACS analysis (**C**) and representative micrographs (**D**) of ROS expression of GEC (green staining) by confocal microscopy studies (magnification ×400, scale bar 50 μm). Nuclei were counterstained in blue by 2.5 μg/mL Hoechst. We performed three experiments with similar results for all the assays, and we performed the statistical analysis by ANOVA with Newmann–Keuls multiple comparison test and the Kolmogorov–Smirnov test. Incubation with cytokines 20 ng/mL TNF -α, 2.5 ng/mL IL-6, plus 50 ng/mL human recombinant C5a protein CKs significantly increased GEC vitality (**A**), inhibited resistance to apoptosis (**B**), and increased oxidative stress (C) in comparison to treatment with vehicle alone (vehicle, * *p* < 0.05 CK vs. vehicle). EV stimulation significantly inhibited these effects (§ *p* < 0.05 CK + EV vs. CK), but not EV were pre-treated with 1 U/mL RNase (# *p* < 0.05 CK + EV RNase vs. CK + EV).

**Figure 8 cells-10-01675-f008:**
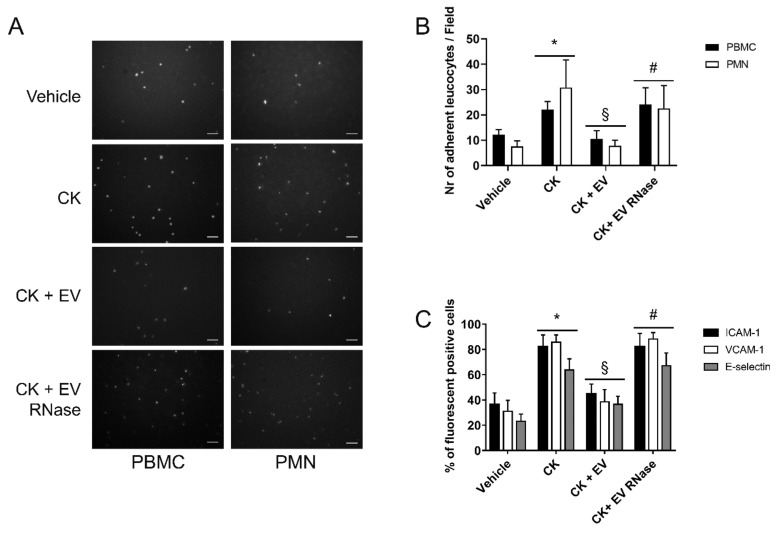
EPC-derived EVs inhibited leukocyte adhesion to GECs. (**A**) Representative images and (**B**) graphs showing the count of adherent PBMCs (black columns) and PMNs (white columns) to GECs. After 24 h of incubation in different culture conditions, GEC monolayers were washed and incubated for 2 h with FITC-labelled PBMC or PMN. We counted adherent FITC-leukocytes in 10 fields/well at x200 magnification under a UV light microscope after fixation (scale bar 50 μm). Data are representative of the average number of adherent cells/field ± 1 SD. We performed three experiments with similar results and statistical analysis by ANOVA with Newmann–Keuls multiple comparison test and Student’s *t*-test. In comparison to vehicle alone, CKs increased the number of adherent FITC-labelled PBMCs or PMNs significantly to GEC monolayers (* *p* < 0.05 CK vs. vehicle). The addiction of 25 μg/mL EVs to medium with CKs, decreased the number of adherent PBMs and PMNs to GECs (§ *p* < 0.05 CK + EV vs. CK) Pre-treatment of EPC-derived EVs with 1 U/mL RNase abrogated this effect (# *p* < 0.05 CK + EV RNase vs. CK + EV). (**C**) FACS analysis of ICAM-1, VCAM-1, and E-selectin in GECs. We expressed the results as the mean of the percentage of positive cells ± 1 SD. We performed the statistical analysis by ANOVA with Newman–Keuls multiple comparison test and Kolmogorov–Smirnov test. In comparison to serum deprivation (vehicle), CKs induced a significant increase in ICAM-1, VCAM-1, and E-selectin expression in GECs (D, * *p* < 0.05 CK vs. vehicle). EVs significantly decreased the expression on the GEC surface of ICAM-1, VCAM-1, and E-selectin (D, § *p* < 0.05 CK + EV vs. CK). By contrast, RNase treatment abrogated these effects induced by EVs (# *p* < 0.05 CK + EV RNase vs. CK + EV). We performed the statistical analysis by ANOVA with Newman–Keuls multiple comparison test and Student’s *t*-test.

**Figure 9 cells-10-01675-f009:**
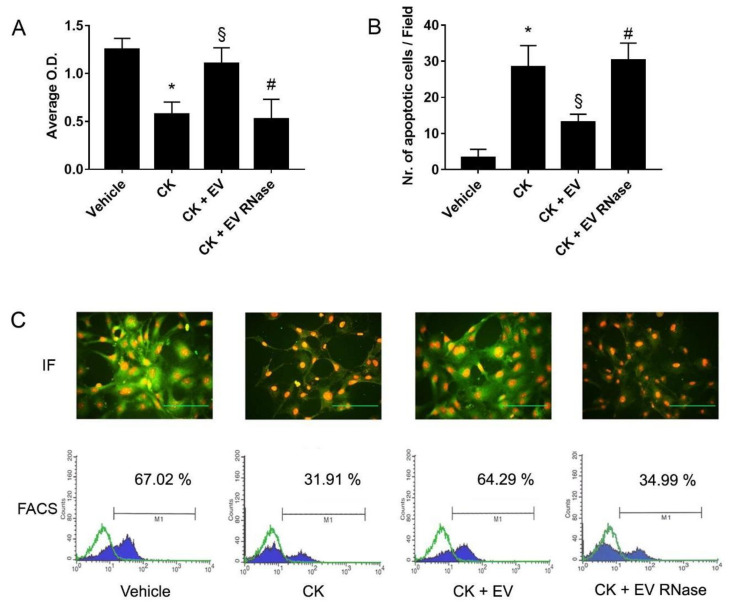
EPC-derived EVs protected podocytes from complement- and cytokine-mediated damage (**A**,**B**). Analysis of podocyte cytotoxicity by XTT assay (**A**) and apoptosis by TUNEL assay (**B**). We performed three experiments with similar results for all the assays and the statistical analysis by ANOVA with Newmann–Keuls multiple comparison test and Student’s *t*-test. For XTT assays, data are expressed as average OD intensity ± 1 S; whereas for TUNEL assays, we report data as the average number of green fluorescent apoptotic cells ± 1 SD. Incubation with CKs significantly increased podocyte apoptosis and inhibited cell viability compared to treatment with vehicle alone (* *p* < 0.05 CK vs. vehicle). EV stimulation significantly inhibited these effects (§ *p* < 0.05 CK + EV vs. CK), but not EV were pre-treated with 1 U/mL RNase (# *p* < 0.05 CK + EV RNase vs. CK + EV). (**C**) Representative micrographs of nephrin expression in podocytes through immunofluorescence studies (IF) and FACS analysis (FACS). We stained nephrin in green for microscope analysis, and we counterstained nuclei with 1 μg/mL propidium iodide; magnification ×400; scale bar 50 μm. For FACS analysis, we compared the staining of nephrin (blue-filled curves) to internal control (green-line curve) represented by appropriate secondary isotype incubation. Stimulation with CKs for 1 h significantly decreased nephrin expression on the cell surface compared to incubation with the vehicle. Podocytes cultured with EPC-derived EVs maintained nephrin expression by inhibiting shedding; RNase pre-treatment of EVs abrogated this effect. For FACS experiments, we performed the Kolmogorov–Smirnov statistical analysis.

**Figure 10 cells-10-01675-f010:**
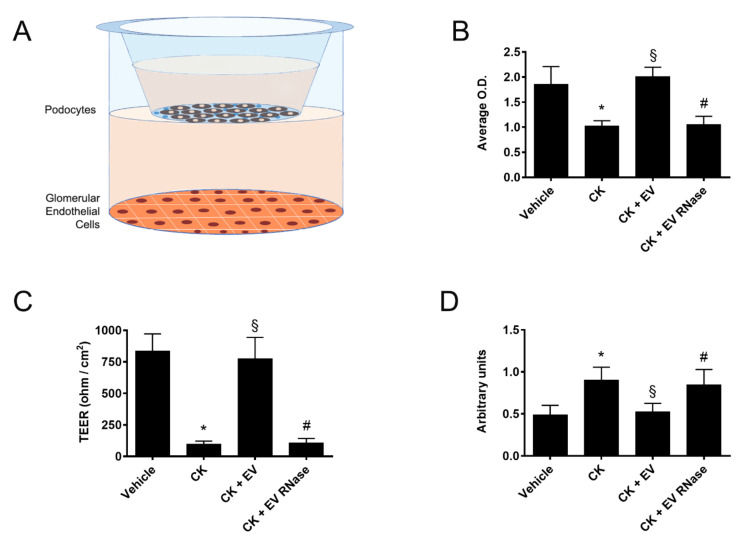
Co-culture model of GECs and podocytes (**A**). Analysis of podocytes cultivated in transwells over GECs in a co-culture model of cytotoxicity by XTT assay (**B**), cell polarity by Trans-Epithelial Electrical Resistance (**C**, TEER), and permeability to Trypan blue-albumin (**D**). We express the data XTT assays as average optical density (OD) intensity ± 1 SD, for TEER results as average ohm/cm^2^ ± 1 SD, and permeability to Trypan blue-albumin as average arbitrary units ± 1 SD. We performed the statistical analysis by ANOVA with Newmann–Keuls multiple comparison test and Student’s *t*-test. If GECs were pre-treated with CKs, podocytes cultivated in transwells in contact with GEC supernatants showed significant loss of vitality (**B**), loss of cell polarity (**C**), and become permeability to Trypan blue-albumin (**D**, * *p* < 0.05 CK vs. vehicle). The addition of 25 μg/mL EPC-derived EVs to GECs significantly protected podocytes from these detrimental effects induced by CKs (§ *p* < 0.05 CK + EV vs. CK). We observed no protective effect of EPC-derived EVs pre-treated with 1 U/mL RNase (# *p* < 0.05 CK + EV RNase vs. CK + EV). We performed three experiments for all the assays with similar results.

## Data Availability

The data presented in this study are available on request from the corresponding author vincenzo.cantaluppi@med.uniupo.it.

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
