# Peer review of "Extracellular Vesicles Derived from Endothelial Progenitor Cells Protect Human Glomerular Endothelial Cells and Podocytes from Complement- and Cytokine-Mediated Injury"

_cells, 2021, doi:10.3390/cells10071675_

Round 1
Reviewer 1 Report
This manuscript struggled with the effect of EPC-derived microvesicles on glomerular cells. The authors conducted many experiments, but because of this, the paper is scattered. Several points must be clarified. Comments are as follows;
Major:
1. Figure 1 have no control. And the authors did not clarified the reason of the gene selection. The replication number of the experiment is not clear. Please clarify, and please keep in mind that the number must be at least three; otherwise, they must conduct qPCR analysis to support reproducibility.
2. In figure 2, the authors did not show increase of angiogenesis. They only showed increase in angiogenic factors. If the authors want to show increase in angiogenesis, they must conduct tube formation assay and cell scratch assay or other morphological assay.
3. This reviewer cannot understand how angiogenesis itself protect glomerular basement membrane. Does angiogenesis occur in glomeruli? Angiogenic factors such as VEGFa are well proven to protect basement membrane or podocytes, but this reviewer does not understand whether angiogenesis itself is renoprotective or not. Moreover, too much VEGF is shown to harmful to podocytes (DOI: 10.1038/ki.2010.64). Can the authors insist the change in Figure 3 really renoprotective?
4. In Figure 5 (although the authors said it's figure 3), they did not show reduced leukocyte adhesion to GECs. They only showed decreased expression of adhesion molecule. Therefore the title of this figure is inappropriate.
5. This reviewer felt the examination of podocytes was abrupt and superficial, and straied from the main topic. If the authors have in vivo data, this experiment is important, but if all of the research is in the cell experiments, this examination about podocytes seems not needed in this research.
6. Considering the points above, this reviewer recommends to i) delete figure 2, 5, and 6, ii) add results which helps mechanistic insights how EVs protect GEC, iii) show decreased leucocyte adhesion (if possible), and iv) perform in vivo analysis since they indicated they could perform in vivo analysis (ref.37) (if possible). These modification must make this manuscript well-focused.
7. Co-culture of GEC and podocytes can reproduce tight junction but not slit diaphagm. Therefore this experiment does not reflect in vivo phenomenum. This point must be clarified as limitation of this study.
8. Some of the paragraph in discussion is a replication of the introduction. Brief description is needed. Other discussion must be re-written according to the change of main figures.
Minor:
1. Judged from figure legends, there are two Figure 3.
2. Threshold p-value must be set before experiment. Please clarify.
3. The discription in "We have previously described that EVs derived from EPC exert a protective effect in Thy1.1 glomerulonephritis on rats by inhibiting antibody- and complement-mediated injury of mesangial cells [31]." is wrong, because the reference number must be 37. The authors must carefully check the reference number.
Author Response
This manuscript struggled with the effect of EPC-derived microvesicles on glomerular cells. The authors conducted many experiments, but because of this, the paper is scattered. Several points must be clarified. Comments are as follows.
We thank the Reviewer for these indications. In our opinion, the experimental procedures that we adopted were necessary to understand better the challenging cross-talk between glomerular endothelial cells and podocytes, the main cells involved in the mechanisms of proteinuria and glomerular damage. We tried to better clarify our choice in the following point-to-point response to the Reviewer.
Major:
1. Figure 1 have no control. And the authors did not clarified the reason of the gene selection. The replication number of the experiment is not clear. Please clarify, and please keep in mind that the number must be at least three; otherwise, they must conduct qPCR analysis to support reproducibility.
Data showed the fold variation of expression of angiogenesis-related genes in GECs stimulated with EVs compared to GECs treated with vehicle alone. The graph in Figure 1 (now moved to Figure 4) is the result of a single experiment on the same panel of genes in PCR array as we performed in our previous papers on EPC-derived EVs and angiogenesis (DOI: 10.1371 / journal.pone.0117530), (DOI: 10.3727 / 096368911X627534). As suggested by the Reviewer, we have confirmed these results by qRT-PCR to support data reproducibility (see new Figure 2S, Appendix A).
2. In figure 2, the authors did not show increase of angiogenesis. They only showed increase in angiogenic factors. If the authors want to show increase in angiogenesis, they must conduct tube formation assay and cell scratch assay or other morphological assay.
The experimental data on GEC angiogenesis (in particular tube formation) were previously shown on page 23 in supplementary data, Appendix A (Figure A2). We have demonstrated the effect of EPC-derived EVs in the classical in vitro angiogenesis assay on Matrigel-coated plates, as shown in representative images and the graphic summarizing the count of capillary-like structures in different experimental conditions. Moreover, we performed proliferation and migration assays by BrdU assay and analysis of migration speed, respectively. We have now moved this figure presenting the role of EPC-derived EVs on GEC angiogenesis in the main figures to make the paper more fluid and understandable (now Figure 3), also following the suggestions of the other Reviewers. Moreover, the results of the present study on EPC EV-induced angiogenesis using human GECs were similar to those observed with other types of endothelial cells in our previously published papers (Deregibus et al. Blood, 2007; Cantaluppi et al. Cell Transplantation 2012; Ranghino et al., Int J Immunopathol Pharmacol 2012; Cantaluppi et al., Kidney Int 2012). We think that these data firmly sustain the role of EPC-derived EVs on GEC angiogenesis.
3. This reviewer cannot understand how angiogenesis itself protect glomerular basement membrane. Does angiogenesis occur in glomeruli? Angiogenic factors such as VEGFa are well proven to protect basement membrane or podocytes, but this reviewer does not understand whether angiogenesis itself is renoprotective or not. Moreover, too much VEGF is shown to harmful to podocytes (DOI: 10.1038/ki.2010.64). Can the authors insist the change in Figure 3 really renoprotective?
We thank the Reviewer for this comment, and we tried to improve the description of the relevance of angiogenesis in the glomerular filtration barrier damage in the course of inflammatory diseases such as glomerulonephritis. Angiogenesis has been demonstrated in different experimental models of glomerulonephritis and in particular in the Thy1.1 model: moreover, a potential role of angiogenesis has been recently described also in human glomerular diseases (Wang L et al., Int J Mol Science 2019; Zhai Y et al., Kidney Blood Press Res 2021). We agree with the Reviewer that Vascular endothelial growth factor A (VEGF) may exert different biologic activities on glomerular cells according to their concentration. However, VEGF (also produced by podocytes in an interesting cellular cross-talk with glomerular endothelial cells) regulates angiogenesis through endothelial cell proliferation and plays an essential role in capillary repair in damaged resident glomerular cells exposed to inflammatory mediators, as already reported in nephritis models (Masuda Y et al., Am J Pathol 2001). Moreover, intussusceptive capillary growth or nonsprouting angiogenesis is involved in the postinjury angiogenesis in the glomeruli through the activities of both glomerular endothelial and mesangial cells (Notoya M et al., Kidney Int 2003). In the clinical context of kidney functional alterations in oncology, VEGF-targeted anti-cancer therapy has been shown to dysregulate the precise molecular cross-talk between podocyte and endothelial cells (Ollero M et al., Nephrol Dial Transpl 2015). Different groups described some of the mediators involved in glomerular angiogenic repair: Ostendorf et al. (J Am Soc Nephrol 2004) confirmed that the VEGF system is of major importance for glomerular endothelial repair in glomerulonephritis and it is significantly affected by nitric oxide (NO) release. Furthermore, the expression of angiopoietins in the glomerulus suggests a mechanism for maintaining the glomerular endothelium and modulation of the actions of glomerular VEGF in both glomerular diseases and recovery from them (Satchell SC et al., J Nephrol 2003). The lack of the matricellular protein thrombospondin-2 (TSP-2) in mice accelerates renal injury: TSP2 is a major endogenous anti-angiogenic and matrix metalloproteinase 2-regulating factor in renal diseases (Daniel C et al., J Am Soc Nephrol 2007). The inhibition of another anti-angiogenic factor such as Endoglin has been shown to promote intussusceptive angiogenesis in experimental nephritis (Hlushchuk R et al., Plos One 2017). Loss of glomerular endothelial cells has been suggested to contribute to the progression of glomerular injury: in this scenario, therapeutic angiogenesis induced by administration of stem cells such as endothelial progenitor cells (EPC) or mesenchymal stromal cells (MSC) may offer an alternative therapeutic option for drug-resistant glomerulonephritis, as observed in other disease models of endothelial injury. The administration of bone marrow-derived angiogenic cells reduced endothelial injury and mesangial activation in anti-Thy1.1 glomerulonephritis: the incorporation into the glomerular endothelial lining and production of angiogenic factor contributed to these protective effects against glomerular injury. (Uchimura H et al., J Am Soc Nephrol 2005). MSC can markedly accelerate glomerular recovery from mesangiolytic damage by releasing paracrine growth factors and not for the differentiation into resident glomerular cell types (Kunter U et al., JASN 2006). Another study showed that bone marrow-derived progenitor cells participate in glomerular endothelial and mesangial cell turnover and contribute to microvascular repair (Rookmaaker MB et al., Am J Pathol 2003). Taking together these data suggest that angiogenesis plays a crucial role in glomerular healing and that stem cells may accelerate this process mainly through the release of paracrine growth factors, including EVs.
4. In Figure 5 (although the authors said it's figure 3), they did not show reduced leukocyte adhesion to GECs. They only showed decreased expression of adhesion molecule. Therefore the title of this figure is inappropriate.
We thank the Reviewer for noticing our error. We have now changed the number of figures from 3 to 5. Since we have added more figures, this is now renamed Figure 8. The inhibition of leukocyte adhesion (PBMC and PMN) is shown in Figure 8B and described in the text on page 10 as follows: “EPC-derived EVs also inhibited PBMC and PMN (Figure 5A) adhesion to GEC monolayers cultured in an inflammatory micro-environment. In addition, EPC-derived EVs significantly down-regulated the protein expression of ICAM-1, VCAM-1, and E-selectin (Figure 5B). Of note, the pretreatment of EVs with RNase abrogated their protective effects (Figures 4 and 5)”. Following Reviewer’s suggestions, we have now added a panel of representative images of leukocyte adhesion to make the paper more understandable together with adhesion molecule expression by GECs (Figure 8A).
5. This reviewer felt the examination of podocytes was abrupt and superficial, and straied from the main topic. If the authors have in vivo data, this experiment is important, but if all of the research is in the cell experiments, this examination about podocytes seems not needed in this research.
We thank the Reviewer for this observation. We think that podocytes are critical cells in maintaining GBM integrity. Indeed, one of the main aims of the present study was to confirm the in vivo data on the Thy1.1 experimental glomerulonephritis model already published by our group (Cantaluppi et al., NDT 2015): in that case, we had observed the reduction of proteinuria and the preservation of renal function in rats treated with EPC-derived EVs. Furthermore, we had analyzed glomerular lesions at the histological level, observing the preservation integrity of glomerular architecture characterized by a decrease of endothelial injury, fusion of podocyte foot processes and mesangial cell activation in Thy1.1 nephritis rats treated by EPC-derived EVs, but not with RNase-treated EVs. Moreover, in the Thy1.1 nephritis model, the EVs co-localized in vivo with synaptopodin (podocyte marker). Treatment with EPC-derived EVs of the Thy1.1 model maintained endothelial and podocyte function. Describing the activity of EPC-derived EVs in an in vitro model on podocytes and the definition of GEC-podocyte cross-talk is a novelty that we believe might be of interest in this field of research.
6. Considering the points above, this reviewer recommends to i) delete figure 2, 5, and 6, ii) add results which helps mechanistic insights how EVs protect GEC, iii) show decreased leucocyte adhesion (if possible), and iv) perform in vivo analysis since they indicated they could perform in vivo analysis (ref.37) (if possible). These modification must make this manuscript well-focused.
i) The PCR array was validated by qRT-PCR. Molecular processes at the level of GECs are essential for understanding the repair mechanisms of GECs following CK damage. For this reason, we have thought to keep these data in the manuscript and added qRT-PCR according to Reviewer’s request.
ii) We have moved the supplementary figures A1 in Appendix A to the main ones. Figure S1 has been added to Appendix A with the characterization of GECs and podocytes by FACS analysis of some L-selectin ligands known to be expressed in the kidney: CD34, podocalyxin and fucosylated residues bound by the molecule Ulex europaeus agglutinin-1 (UEA-1). In vitro cell characterization demonstrated that GECs express CD34, podocalyxin and fucosylated residues, whereas podocytes only express podocalyxin. All these molecules can be involved in L-selectin-mediated EV internalization within target glomerular cells. This point is now better discussed in the text.
iii) We have already shown leukocyte adhesion in Figure 5A in the first version of the manuscript. We have now added representative images of leukocyte adhesion and renamed the figure (Figure 8), also following the other Reviewers' comments.
iv) We have already demonstrated the effect of EPC-derived EVs in an in vivo model of Thy1.1 glomerulonephritis (Cantaluppi et al., NDT 2015). In this model, we observed the reduction of proteinuria and the preservation of renal function in rats treated with EPC-derived EVs but not with RNase-treated EVs. Furthermore, we analyzed glomerular lesions at the histological level, showing the integrity of glomerular architecture characterized by a decrease of endothelial injury, fusion of podocyte foot processes and mesangial cell activation by electron microscopy analysis. Since we have previously focused on the protective effect of EPC-derived EVs on complement-mediated mesangial cell damage in vitro, we have herein focused on experiments on injured GECs and podocytes to describe further EPC-derived effects of EVs on glomerular healing and repair. Therefore, this work mainly aims to evaluate the in vitro effect on cell lines and the co-culture model GEC-podocytes following our previous in vivo
7. Co-culture of GEC and podocytes can reproduce tight junction but not slit diaphagm. Therefore this experiment does not reflect in vivo phenomenum. This point must be clarified as limitation of this study.
We have added micrograph A to figure 10 to understand better the co-culture model we use. We kept the GECs and podocytes in two separate compartments to mimic the in vivo situation. We first incubated the wells with GECs without podocytes in the presence or not of EPC-derived EVs for 24 hours. Then, we washed and added a new medium without any stimuli. Then, podocyte cells cultured in transwells were added to the wells of GECs for another 24 hours. The transwell contains pores that allow the passage of molecules but not of cells (4 micrometres in diameter). In this way, the podocytes just contacted what was released by the GECs in the new medium. We recognize that this is a limitation of the experimental procedure: however, it represents a good even not optimal manner to evaluate GEC-podocyte cross-talk and their influence on neighbouring cells. This point is now discussed in more details in the text.
8. Some of the paragraph in discussion is a replication of the introduction. Brief description is needed. Other discussion must be re-written according to the change of main figures.
We have changed the discussion considering this comment.
Minor:
1. Judged from figure legends, there are two Figure 3.
Thanks for your note. Please, see our answer in Major comment # 4.
2. Threshold p-value must be set before experiment. Please clarify.
In all the legends of the figures, the threshold p-value was specified. However, to clarify more on page 8 in statistical methods, we have added: “Significance level for all tests was set at p < 0.05”.
3. The discription in "We have previously described that EVs derived from EPC exert a protective effect in Thy1.1 glomerulonephritis on rats by inhibiting antibody- and complement-mediated injury of mesangial cells [31]." is wrong, because the reference number must be 37. The authors must carefully check the reference number
Thanks for noticing this error. We have changed the numbering of reference from 31 to 38, considered we have added reference 30.
Reviewer 2 Report
It is a scientific paper of great interest where the authors suggest a very attractive therapeutic alternative to glomerular damage.
1. First of all, I would like to add two comments on the text format:
1.1 Latin words such as in vivo or in vitro must be italicized or quoted
1.2 The caption read in figure 6 must be checked ( subparagraphs A and B)
2. In experimental methodology I would like to know why authors decided to use each of the statistical tests in each case
3. About the results, I would like to know if the authors have conducted any experiments or have any hypotheses about how the process of internalization of EVs could occur, apart from being a process in which L-selectin is involved? What concentration of EVs do you use? Have you conducted experiments to evaluate response based on the amount of EVs ?
4. Have you planned or conducted this study on an experimental model of glomerulonephritis in rodents to observe histologically what happens to gbm and also to monitor how permeabilit would recover after EVs internalization?
Author Response
It is a scientific paper of great interest where the authors suggest a very attractive therapeutic alternative to glomerular damage.
- First of all, I would like to add two comments on the text format:
1.1 Latin words such as in vivo or in vitro must be italicized or quoted.
We have italicized Latin words in all the manuscript. Thanks for the suggestion.
1.2 The caption read in figure 6 must be checked ( subparagraphs A and B).
We have corrected it in the text. We have changed the XTT assay and TUNEL assay description in this figure (now re-named Figure 9).
- In experimental methodology I would like to know why authors decided to use each of the statistical tests in each case.
We performed Statistical analyses by one-way ANOVA followed by the Student-Newman-Keuls post hoc test after three different experiments because this test is quite robust to violations of normality and the assumptions of this test essentially are the same as for an independent group's t-test: normality, homogeneity of variance, and independent observations. For FACS analysis, because it describes a non-parametric distribution, the Kolmogorov–Smirnov test is a valid statistical solution for comparing distributions that have been recommended for flow cytometric histogram analysis (Lampariello, Cytometry. 2000 Mar 1;39(3):179-88. doi:10.1002/(SICI)1097-0320(20000301)39:3<179::AID-CYTO2>3.0.CO;2-I). Moreover, we adopted the same statistical tests for our previous in vitro studies using EPC-derived EVs (Cantaluppi et al., Kidney Int 2012, Cantaluppi et al., NDT 2015). - About the results, I would like to know if the authors have conducted any experiments or have any hypotheses about how the process of internalization of EVs could occur, apart from being a process in which L-selectin is involved? What concentration of EVs do you use? Have you conducted experiments to evaluate response based on the amount of EVs?
We thank the reviewer for this comment. We have now added the characterization of our cell lines by evaluating the expression of some L-selectin ligands (Figure S1, Appendix A). We hypothesized that the internalization in GECs takes place mainly through CD34 and podocalyxin, whereas in podocytes only through podocalyxin. Furthermore, GECs express the fucosylated residues that allow L-selectin binding. This follows our previous in vitro studies using EPCs (Biancone et al., J Immunol 2004) and EPC-derived EVs (Deregibus et al., Blood 2007; Cantaluppi et al., Kidney Int 2012, Cantaluppi et al., NDT 2015). We first evaluated the optimal EV dose and timing of incubation (Figure S3, Appendix A): then, we conducted the experiments with a dose of 25 micrograms/mL, according to our previous data. Moreover, the quantification of EPC-derived EVs preparations was performed both by Bradford and Nanosight methods: one microgram of preparation of EVs contains approximately 109 microparticles, and we then used this particle concentration determined by Nanosight in all experimental settings also following our above-mentioned previous publications in the field (Deregibus et al., Blood 2007; Cantaluppi et al., Kidney Int 2012, Cantaluppi et al., NDT 2015). We can not exclude other mechanisms nno dependent on L-selectin involved in EV internalization. However, EV seems to use a mechanism of cell adhesion and internalization similar to that adopted from the cells of origin (EPC: see also Biancone et al. J Immunol 2004). These points are now better discussed. - Have you planned or conducted this study on an experimental model of glomerulonephritis in rodents to observe histologically what happens to gbm and also to monitor how permeability would recover after EVs internalization?
We have previously performed this type of experiments in the Thy1.1 experimental glomerulonephritis model (Cantaluppi et al., NDT 2015). In this model, we observed the reduction of proteinuria and the preservation of renal function in rats treated with EPC-derived EVs but not with RNase-treated EVs. Furthermore, we analyzed glomerular lesions at the histological level, showing the integrity of glomerular architecture characterized by a decrease of endothelial injury, fusion of podocyte foot processes and mesangial cell activation by electron microscopy analysis. Since we have previously focused on the protective effect of EPC-derived EVs on complement-mediated mesangial cell damage in vitro, we have herein focused on experiments on injured GECs and podocytes further to describe EPC-derived EV effects on glomerular healing and repair.
Reviewer 3 Report
The paper described that EPC-derived EVs protect GEC and podocytes from complement- and cytokine-induced damage by changing the expression of the genes involved in GEC angiogenesis, increasing the secretion of VEGF-A and HGF, and reducing the production of ROS and apoptotic cells. The authors suggested that RNA transfer from EVs to damaged glomerular cells is important. The results are interesting, and potentially important to develop the therapeutic agents for drug-resistant glomerulonephritis. However, there are issues that need to be addressed.
Major points:
(1) Throughout the manuscript, the authors just mentioned the significant differences without showing how much percentage/how many times has/have changed though most of the data were quantified, which should be included.
(2) The authors proposed the mechanism for internalization of EPC-derived EVs in L-selectin-dependent manner. However, they did not show whether L-selectin and its ligand are expressed in EVs, GECs, and podocytes.
(3) To show the biological significance of RNA transfer, the authors treated EVs with RNase, but the confirmation of RNase-treatment that shows the degradation of mRNA and microRNA was not shown. Furthermore, whether the treatment with RNase affects the internalization of EVs to glomerular cells was not shown.
(4) The authors should discuss how EVs is involved in the inhibition of nephrin shedding.
(5) Since the quality of the table in Figure 2B is low, it is hard to see. The authors must improve it.
(6) About Figure A2D in Appendix A, without symbols that indicate the different samples, the lines in the graph cannot be distinguished. For reader’s understanding, the reviewer recommends that the authors use the lines with symbols to show the different samples.
Minor points:
(1) In Materials and Methods, “one h” and “1 h” should be unified. In page 5, line 2, “we” should be “We”.
(2) Although the authors mentioned “EGF” in page 6, line 10 from the bottom and in the legend of Figure 1, “EGF” is not in the graph of Figure 1.
(3) In page 6, line 1 from the bottom, the authors mentioned “miR324-5P”, but “miR-324-3P” is in Figure 2A and 2B. The results and legend should be revised. Moreover, “miR-485-3P” is not described in the results and legend of Figure 2. “BAI1” and “FIGF” in the legend of Figure 2 are not in Figure 2A and 2B.
(4) In Figure 3C and its legend, “VEGF” should be “VEGF-A”.
(5) In page 12, line 1, “Figure 3” should be “Figure 5”.
Author Response
The paper described that EPC-derived EVs protect GEC and podocytes from complement- and cytokine-induced damage by changing the expression of the genes involved in GEC angiogenesis, increasing the secretion of VEGF-A and HGF, and reducing the production of ROS and apoptotic cells. The authors suggested that RNA transfer from EVs to damaged glomerular cells is important. The results are interesting, and potentially important to develop the therapeutic agents for drug-resistant glomerulonephritis. However, there are issues that need to be addressed.
Major points:
(1) Throughout the manuscript, the authors just mentioned the significant differences without showing how much percentage/how many times has/have changed though most of the data were quantified, which should be included.
As suggested by the Reviewer, in the results, we have now reported averages + 1 standard deviation (SD) of the bar graphs shown in the figures. We have also changed the text accordingly, indicating the number of percentages and absolute number variations when data allowed it.
(2) The authors proposed the mechanism for internalization of EPC-derived EVs in L-selectin-dependent manner. However, they did not show whether L-selectin and its ligand are expressed in EVs, GECs, and podocytes.
We thank the Reviewer for underlying this important point, and we have now added Figure S1 in Appendix A, with the characterization of GECs and podocytes by FACS analysis of some of the different L-selectin ligands known to be expressed in the kidney: CD34, podocalyxin and fucosylated residues bound by the molecule Ulex europaeus agglutinin-1 (UEA-1). Moreover, in vitro cell characterization demonstrated that GECs express CD34, podocalyxin and fucosylated residues, whereas podocytes express only podocalyxin.
(3) To show the biological significance of RNA transfer, the authors treated EVs with RNase, but the confirmation of RNase-treatment that shows the degradation of mRNA and microRNA was not shown. Furthermore, whether the treatment with RNase affects the internalization of EVs to glomerular cells was not shown.
We thank the Reviewer for this suggestion: as reported in the discussion and the references, we have previously demonstrated that pre-treatment with non-physiological concentrations of RNase can limit the biological effects of EPC-derived EVs on angiogenesis (Deregibus et al. Blood, 2007, Cantaluppi et al. Cell Transplantation 2012, Ranghino et al., Int J Immunopathol Pharmacol 2012) and kidney tubular and glomerular injury (Cantaluppi et al., Kidney Int 2012, Cantaluppi et al., NDT 2015). These effects are mainly ascribed to the degradation of total RNA and, in particular, the small RNase induced by this enzyme. Moreover, RNA degradation seems to be facilitated by the use of non-physiological doses of RNase and by the sudden change of temperature at 37°C after the ultracentrifugation step at 4°C, without any significant alteration of surface protein expression essential for EV internalization within target cells. Basing on these considerations and following Reviewer’s suggestions, we now show in Figure 2 the effect of RNase on EPC-derived EVs in terms of number, size and content of RNA (total RNAs and small RNAs) and protein (L-selectin and integrins evaluated in the internalization assay in Figure 1C) variation. RNase treatment did not alter EV concentration (median and mode of RNase-treated and untreated EVs are comparable): taking together, as already published in our previous studies, RNase treatment degraded total RNA but did not modulate EV protein content.
(4) The authors should discuss how EVs is involved in the inhibition of nephrin shedding.
From a methodological point of view, we treated podocytes for 24 hours with EPC-derived EVs, EPC-derived EVs pre-treated with RNase or with vehicle alone to perform this selected experiment. At the end of different incubations, we stimulated cells with inflammatory mediators for 1 hour, similar to the experiments conducted in a previous paper (DOI: 10.2337/diabetes.52.4.1023). Of note, we did not observe any significant difference in nephrin expression among different experimental conditions before the incubation with CK. We have now described this experiment more accurately within the text. The mechanisms of EV-induced inhibition of nephrin shedding may be multifactorial and not easy to describe: our previous in vivo experiments (Cantaluppi et al., NDT 2015) demonstrated that EPC-derived EVs could be internalized in both glomerular endothelial cells and podocytes in Thy1.1 experimental model of glomerulonephritis: moreover, EV treatment preserved the expression of both endothelial and podocyte markers in Thy1.1 Ab-treated rats. Nephrin is a protein of the podocyte slit diaphragm essential for the maintenance of glomerular permselectivity and lost in different pathologic conditions, in particular in the presence of nephrotic syndrome. Our group have previously shown that nephrin shedding occurs in response to different stimuli, including drugs (e.g. mTOR inhibitors) and inflammatory mediators (Biancone et al., Am J Transpl 2010; Mariano et al., Crit Care 2008). On this basis, we can hypothesize a direct effect of EVs on podocytes able to inhibit nephrin shedding. However, since most EV-related biological activities are related to RNA transfer, the preservation of podocyte viability and modulation of nephrin gene expression should also be considered. Last, we have underlined the relevance of GEC-podocyte cross-talk: we herein demonstrated the EV-induced triggering of GEC angiogenesis with the possible increased release of trophic factors for podocytes. All these mechanisms are now better discussed.
(5) Since the quality of the table in Figure 2B is low, it is hard to see. The authors must improve it.
We have modified the organization of the micrographs to allow a better understanding of data for the reader and to improve quality. The figure has now been re-named Figure 5.
(6) About Figure A2D in Appendix A, without symbols that indicate the different samples, the lines in the graph cannot be distinguished. For reader’s understanding, the reviewer recommends that the authors use the lines with symbols to show the different samples.
Data are now presented using a colour graph. We have added symbols for each experimental condition. Also, following the suggestion of the other Reviewers, this figure has now been moved from the supplementary material in Appendix A to the main figures of the text as Figure 3.
Minor points: Thanks for noticing these mistakes. We have now corrected them.
(1) In Materials and Methods, “one h” and “1 h” should be unified. In page 5, line 2, “we” should be “We”.
We have unified the text using "1 hour" and corrected the lowercase letter on page 5, line 2.
(2) Although the authors mentioned “EGF” in page 6, line 10 from the bottom and in the legend of Figure 1, “EGF” is not in the graph of Figure 1.
We have deleted it from the text; we apologize for this mistake.
(3) In page 6, line 1 from the bottom, the authors mentioned “miR324-5P”, but “miR-324-3P” is in Figure 2A and 2B. The results and legend should be revised. Moreover, “miR-485-3P” is not described in the results and legend of Figure 2. “BAI1” and “FIGF” in the legend of Figure 2 are not in Figure 2A and 2B.
We have corrected these errors in the legend of Figure 5 and the results.
(4) In Figure 3C and its legend, “VEGF” should be “VEGF-A”.
We made corrections in Figure 6 and related legend.
(5) In page 12, line 1, “Figure 3” should be “Figure 5”.
Thanks for having noticed it, and we apologize for the mistake. In the new version of the manuscript, we have added some main figures, and this is now re-named Figure 8.
Round 2
Reviewer 1 Report
The authors performed enough additional experiments and rewrote the manuscript well. Now this manuscript seems appropriate to be published.
Reviewer 3 Report
There is no criticism because the authors have addressed all the reviewer’s comments appropriately.